# CLAVATA signalling shapes barley inflorescence by controlling activity and determinacy of shoot meristem and rachilla

Isaia Vardanega [1], Jan Eric Maika [1], Edgar Demesa-Arevalo [1,2], Tianyu Lan [3], Gwendolyn K. Kirschner [1,2], Jafargholi Imani [4], Ivan F. Acosta [5], Katarzyna Makowska[6], Götz Hensel [6], Thilanka Ranaweera [7,8,9], Shin-Han Shiu [7,8,9], Thorsten Schnurbusch [10,11], Maria von Korff[2,3] & Rüdiger Simon [1,2] ✉

The large variety of inflorescence architectures evolved in grasses depends on shape, longevity and determinacy of meristems directing growth of the main and lateral axes. The CLAVATA pathway is known to regulate meristem size and inflorescence architecture in grasses. However, how individual meristem activities are determined and integrated to generate specific inflorescences is not yet understood. We found that activity of distinct meristems in the barley inflorescence is controlled by a signalling pathway comprising the receptor-like kinase *Hordeum vulgare* CLAVATA1 (HvCLV1) and the secreted CLAVATA3/EMBRYO-SURROUNDING REGION RELATED (CLE)-family peptide FON2-LIKE CLE PROTEIN1 (HvFCP1). HvFCP1 and HvCLV1 interact to promote spikelet formation, but restrict inflorescence meristem and rachilla proliferation. *Hvfcp1* or *Hvclv1* mutants generate additional rows of spikelets and supernumerary florets from extended rachilla activity. *HvFCP1/HvCLV1* signalling coordinates meristem activity through regulation of trehalose-6-phosphate levels. Our discoveries outline a path to engineer inflorescence architecture via specific regulation of distinct meristem activities.

The *Poaceae* family (*Gramineae* or grasses) displays a large variety of inflorescence architectures that evolved from an ancestral compound spike with panicle-like branches on a main inflorescence axis (rachis)[1]. The several branching orders characteristic of panicles gradually simplified during evolution, generating the variety of current grass inflorescences[2]. Distinct grass inflorescence architectures depend on the initiation and placement of new meristems on the flanks of the shoot apical meristem (SAM), on meristem size, longevity, identity and determinacy[3]. During the vegetative phase, the cereal SAM generates leaf primordia in a distichous pattern. In response to external and internal signals, the SAM converts into an inflorescence meristem (IM), which can initiate spikelets directly on the rachis in *Triticeae*,

[1]Institute of Developmental Genetics, Heinrich-Heine University, Düsseldorf, Germany. [2]CEPLAS, Center of Excellence in Plant Sciences, Heinrich-Heine University, Düsseldorf, Germany. [3]Institute of Plant Genetics, Heinrich-Heine University, Düsseldorf, Germany. [4]Institute of Phytopathology, Justus Liebig University, Giessen, Germany. [5]Max Planck Institute for Plant Breeding Research, Cologne, Germany. [6]Centre for Plant Genome Engineering, Institute of Plant Biochemistry, Heinrich-Heine University, Düsseldorf, Germany. [7]Department of Plant Biology, Michigan State University, East Lansing, MI, USA. [8]DOE-Great Lake Bioenergy Research Center, Michigan State University, East Lansing, MI, USA. [9]Department of Computational Mathematics, Science, and Engineering, Michigan State University, East Lansing, MI, USA. [10]Leibniz Institute of Plant Genetics and Crop Plant Research (IPK), Gatersleben, Germany. [11]Institute of Agricultural and Nutritional Sciences, Martin Luther University Halle-Wittenberg, Halle, Germany. ✉e-mail: ruediger.simon@hhu.de

including barley (*Hordeum vulgare L.*) and wheat (*Triticum ssp.*), or form primary and secondary branches in *Oryzeae* (rice) and *Andropogoneae* (maize and sorghum)[4].

Spikelet meristems (SM) give at first rise to two small modified bracts (the glumes), and later to a variable number of floret(s) that develop from floret meristems (FMs) on a short axis called rachilla. Florets carry the leaf-like lemma and palea, which enclose modified petals called lodicules, and the sex organs, stamen and carpel[5,6].

In barley, the IM remains indeterminate and generates triple spikelet meristems (TSM) on its flanks until developmentally programmed pre-anthesis tip degeneration causes IM senescence and death[5,7,8]. TSM splits into a central spikelet meristem (CSM), flanked by two lateral spikelet meristems (LSM). In two-rowed genotypes, lateral spikelets remain sterile and arrest before floret organs are fully developed[9]. Each SM further divides into the vestigial rachilla primordium (RP), an abaxial floret meristem (FM) and a subtending lemma primordium (LEP)[10].

Inflorescence branching is suppressed by the transcription factors COMPOSITUM1 (COM1), COM2, VRS4 and HvMADS1, while INTERMEDIUM-m restricts floret number per spikelet and maintains indeterminacy of the IM[11–15]. In maize, inflorescence branching is controlled by the RAMOSA pathway, comprising RAMOSA3 (RA3), a Trehalose-6-Phosphate Phosphatase that putatively regulates branching by a sugar signal that moves into axillary meristems[16]. Importantly, even closely related grasses, such as the temperate cereals wheat and barley, differ in their inflorescence architecture because of differences in meristem behaviour and determinacy. In wheat (*Triticum aestivum* L.), inflorescence growth arrests with differentiation of its determinate IM into an SM, while its rachilla remains indeterminate, enabling the formation of up to 12 florets[2,5,10]. These examples illustrate how differential regulation of meristem activities finally impacts inflorescence architecture. However, the regulatory networks, feedback regulations and external inputs that coordinate the size, longevity and determinacy of different meristem types in the inflorescence are still unknown.

In Arabidopsis, the activity of shoot and floral meristems depends on the WUSCHEL transcription factor, which moves from a deeper meristem region to the stem cell zone to promote stem cell maintenance by suppressing auxin response factors[17,18]. WUSCHEL promotes expression of the CLAVATA3/EMBRYO SURROUNDING REGION (CLE) family peptide CLAVATA3 (CLV3), which is secreted from stem cells and interacts with leucine-rich-repeat receptor kinases (LRR-RLKs) of the CLAVATA1 (CLV1) and coreceptors of the CIK family to repress WUSCHEL, thereby providing a negative feedback signal. The CLE40 peptide, which is closely related to CLV3, acts from the meristem periphery through the CLV1-related RLK BARELY ANY MERISTEM 1 (BAM1) to impact meristem activity[18]. CLV-related signalling pathways were found to regulate diverse meristem activities, including root and cambial meristems[19], and CLE/CLV1-family signalling has also been shown to affect the size of different meristem types in grass species. In rice, the LRR-RLK FLORAL ORGAN NUMBER1 (FON1) and the CLE peptide FLORAL ORGAN NUMBER2 (FON2) restrict the sizes of FMs, IM and the number of primary branches[20,21], while FON2-LIKE CLE PROTEIN1 (FCP1), which is highly conserved between cereal grasses, likely plays an antagonistic role and promotes maintenance of the vegetative SAM and root apical meristem[22–24]. Overexpression of FCP1 in WT and *fon1-5* null mutant showed similar meristem phenotypes in rice, suggesting that their function relies on independent pathways, while the insensitivity of *fon1-5* to OsFON2 overexpression indicated their possible interaction[25]. The maize LRR-RLK THICK TASSEL DWARF1 (TD1) and CLE7 confine the diameter of both tassel and ear inflorescence meristems. In *td1* mutants, an enlarged IM initiates disorganised supernumerary rows of Spikelet Pair Meristems, that sometimes develop additional SMs[26,27]. In parallel, ZmFCP1 signalling suppresses stem cell proliferation in the ear inflorescence meristem, which enlarged in *Zmfcp1* mutants[23,27]. Mutants in SvFON2, the CLE

peptide ortholog of OsFON2 and ZmCLE7 in *Setaria viridis*, developed larger inflorescence meristems, similar to mutants in maize and rice[28]. However, while *Svfon2* and Zm*FCP1CR-pro2* showed elongated branch meristems, no differences in the tassel branches were detected in *Zmcle7*[27,28]. No differences in vegetative traits like tiller number or height were detected in *Svfon2* or *Zmcle7*, while *Osfon4 (Osfon2 allele)* presented thick culms and a reduced number of tillers[29]. Additionally, SvFON2 plays a role in flowering time, with mutants reaching the heading stage earlier than WT[28].

The development of cereal inflorescence architectures requires close coordination of meristem ontogenies to ensure the generation of different organs in the right number, position and time. Orthologs of CLAVATA components were shown to regulate the size of different meristem types within grass inflorescences, ultimately influencing their architecture; however, how individual meristem activities are coordinated and integrated to generate specific inflorescences is not yet understood. Here, we investigate how CLV-related signalling pathways may contribute to this process in barley. We identify the barley *HvCLV1* and *HvFCP1* genes and show that they act in joint but also separate signalling pathways to impact multiple aspects of inflorescence and spikelet meristem development, affecting plant architecture and flowering time. Our findings extend the roles for CLV signalling, from feedback signalling in stem cell homoeostasis to coordination of meristem shape, organ formation and determinacy in cereal inflorescences.

## Results

To identify CLV1-related RLKs from barley, we analysed the phylogeny for all protein kinase sequences from two dicotyledons (*Arabidopsis thaliana*, *Solanum lycopersicum*) and four gramineous species belonging to the *Poaceae* family (*Zea mays*, *Oryza sativa japonica*, *Triticum turgidum* and *Hordeum vulgare*) (Supplementary Data 1). Within the clade comprising AtCLV1, we identified six closely related genes from barley. HORVU.MOREX.r3.7HG0747230 represented the closest ortholog of AtCLV1 in *Hordeum vulgare* and was named *HvCLV1*. *HvCLV1* grouped with the maize and rice orthologs *ZmTD1* and *OsFON1*, the other five genes in the clade were more closely related to *AtBAM1 to 3* and named *HvBAM1* to *5* (Supplementary Fig. 1a). *HvCLV1* encodes an LRR-RLK protein of 1015 amino acids, comprising an intracellular kinase domain and 20 extracellular Leucin Rich Repeats (LRRs), similar to the closely related ZmTD1 and OsFON1, and AtCLV1 with 21 LRRs (Fig. 1a, Supplementary Fig. 1a). *HvCLV1* expression was analysed using single-molecule RNA fluorescent in situ hybridisation (smRNA-FISH, Molecular Cartography™, Resolve Biosciences) on sectioned developing barley apices during vegetative (Fig. 1b–d) and reproductive stages (Fig. 1e–m, Supplementary Fig. 1b). HvCLV1-protein localisation was analysed using the translational reporter line *pHvCLV1:HvCLV1-mVenus*, which expresses the HvCLV1 protein with the fluorophore mVenus fused C-terminally to the cytoplasmic kinase domain and functionally complements a *Hvclv1* mutant (see below, Supplementary Fig. 4). We used the Waddington scale to define stages (Waddington stage, W) of barley development[30]. During vegetative (W1) and reproductive development (W3.5) (Fig. 1b, e), *HvCLV1* is expressed mostly in the three outer cell layers in the apical meristem (Fig. 1c, f), and along spikelet development in RP, FM, lemma primordium and floral organs (Fig. 1h, l). *HvCLV1* mRNA signals were also present in leaf primordia (Fig. 1c), and HvCLV1 proteins were generally detected in similar patterns (Fig. 1d, g, i, m). Even though HvCLV1 is broadly expressed in the plasma membrane of cells comprising the spikelet organs, longitudinal sectioning of the developing spikelet showed an enhanced HvCLV1 protein internalisation in the cytoplasmic structures of RP cells compared to the adjacent FM (Fig. 1j, k, Supplementary Fig. 1c, d). This observation might reflect an enhanced de novo synthesis and intracellular trafficking of HvCLV1 in these cells, or turnover after signalling of HvCLV1 receptors in the RP[31].

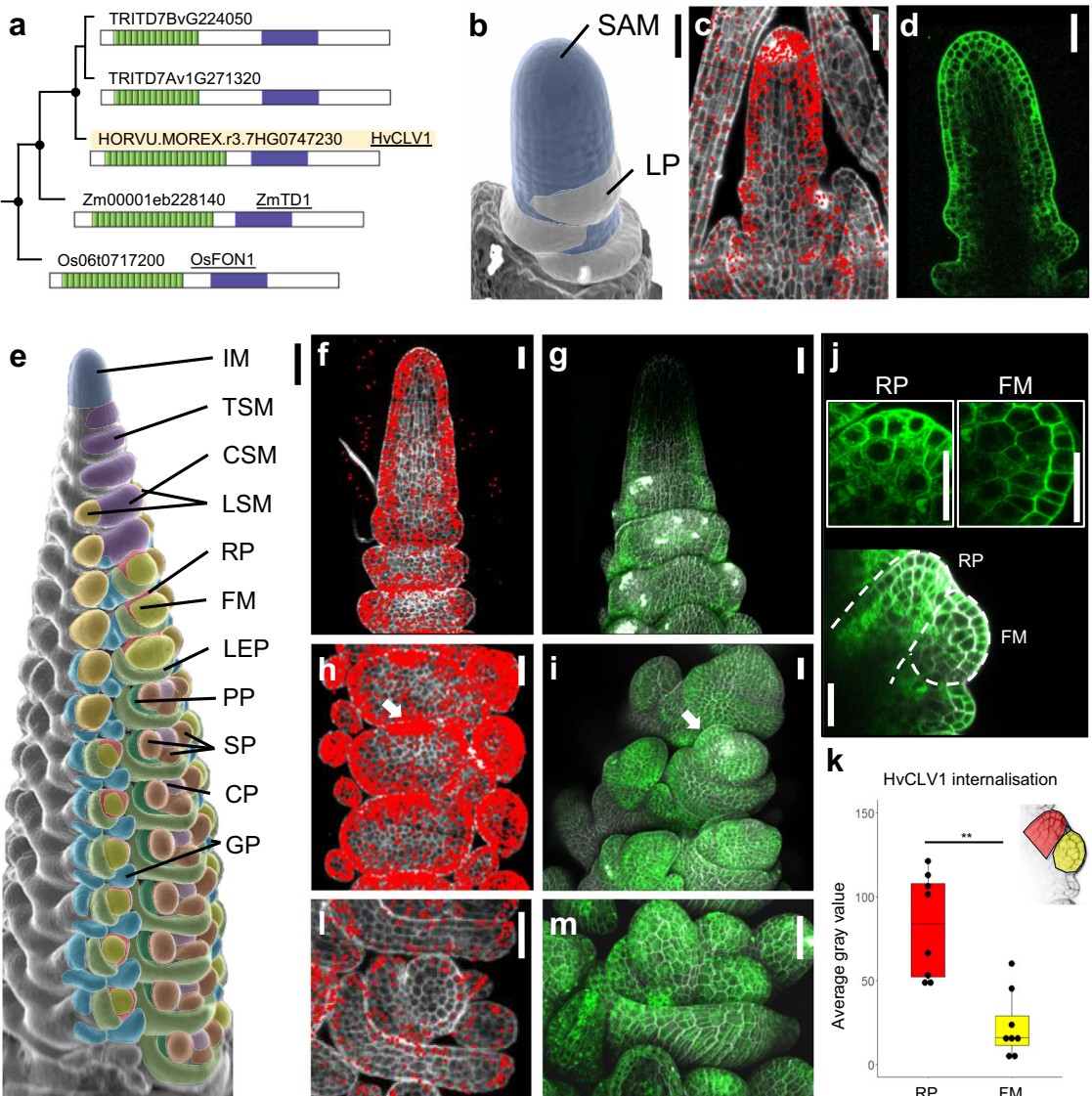

**Fig. 1 | Identification, protein localisation and expression pattern of *HvCLV1* in different meristems comprising the barley inflorescence. a** Maximum likelihood tree of the HvCLV1 subclade. Black dots indicate nodes with bootstrap values higher than 80. Gene identifiers are shown next to a schematic representation of protein structures. Kinase domain in purple and LRRs in green. HvCLV1 is highlighted in light orange. **b** SEM picture of barley vegetative meristem. Colour code: shoot apical meristem (SAM) in blue, leaf primordia (LP) in white. **c** smRNA-FISH detection of *HvCLV1* transcripts (red dots) at the vegetative stage, calcfluor stained cell wall in grey. **d** HvCLV1 protein localisation in a central longitudinal section of the SAM at vegetative stage, HvCLV1 proteins tagged with mVenus in green. **e** *Hordeum vulgare* inflorescence *cv*. Golden Promise Fast, W3.5. Colour code: inflorescence meristem (IM) in blue, triple spikelet meristem (TSM) and central spikelet meristem (CSM) in purple, lateral spikelet meristem (LSM) in orange, rachilla primordium (RP) in red, floret meristem (FM) in yellow, lemma primordium (LEP) in light green, palea primordium (PP) in dark green, stamen

primordia (SP) in brown, carpel primordium (CP) in pink and glumes primordia (GP) in cyan. **f**, **g** Transcripts and proteins localisation of HvCLV1 in the IM and TSMs, **h**, **i** in spikelet primordia at the FM initiation stage. The white arrows indicate the RP. **j** Central longitudinal section of SM. Segmented lines indicate RP and FM. The close-up pictures show HvCLV1 proteins internalised in the vacuole in the RP and HvCLV1 proteins localised on the plasma membrane in the FM. **k** Quantification of HvCLV1 protein internalisation in RP and FM. Dots represent single measurements, and asterisks indicate the significant difference between samples using a two-sided Wilcoxon rank sum exact test. $n = 8$ central spikelets over $n = 2$ independent experiments. Boxplots: median (centre line); upper and lower quartiles (box limits); 1.5x interquartile range (whiskers). **l**, **m** HvCLV1 transcripts and proteins localisation in stamens and carpel primordia. *HvCLV1* transcripts are in red and HvCLV1 proteins in green. Scale bars = 50 μm, in **e** = 100 μm. Statistics: ns non-significant (*p*-value > 0.05); *(*p*-value < 0.05); **(*p*-value < 0.01); ***(*p*-value < 0.001).

For a better functional understanding of the role of barley HvCLV1, we generated *Hvclv1* mutants by CRISPR-Cas9. Three independent alleles, *Hvclv1-1* to −3, which likely represent loss-of-function mutants, showed closely related phenotypes (Supplementary Fig. 2a, Supplementary Notes 1). All *Hvclv1* mutants were semi-dwarfs (Fig. 2a), with shorter stems, spikes and fewer internodes than WT (Supplementary Fig. 2b–d). The number of tillers was reduced in *Hvclv1-1* and −2, but not significantly different from WT in *Hvclv1-3* (Supplementary Fig. 2e). All *Hvclv1* mutant alleles also developed

fewer, smaller, and lighter grains than WT (Supplementary Fig. 2f–k). Furthermore, a variable proportion of the *Hvclv1-1* spikes formed additional, ectopic rows of spikelets in a non-distichous phyllotaxis (crowned spikes) (Fig. 2b–d), or carried multi-floret spikelets with two or three florets, separate embryos and endosperms enclosed by partially fused lemmas (Fig. 2e–g). These phenotypes were also observed in *Hvclv1* plants grown in semi-field-like conditions in Germany, between March to the end of July 2023, but not in WT (Supplementary Fig. 3a, b). Time-course analysis showed

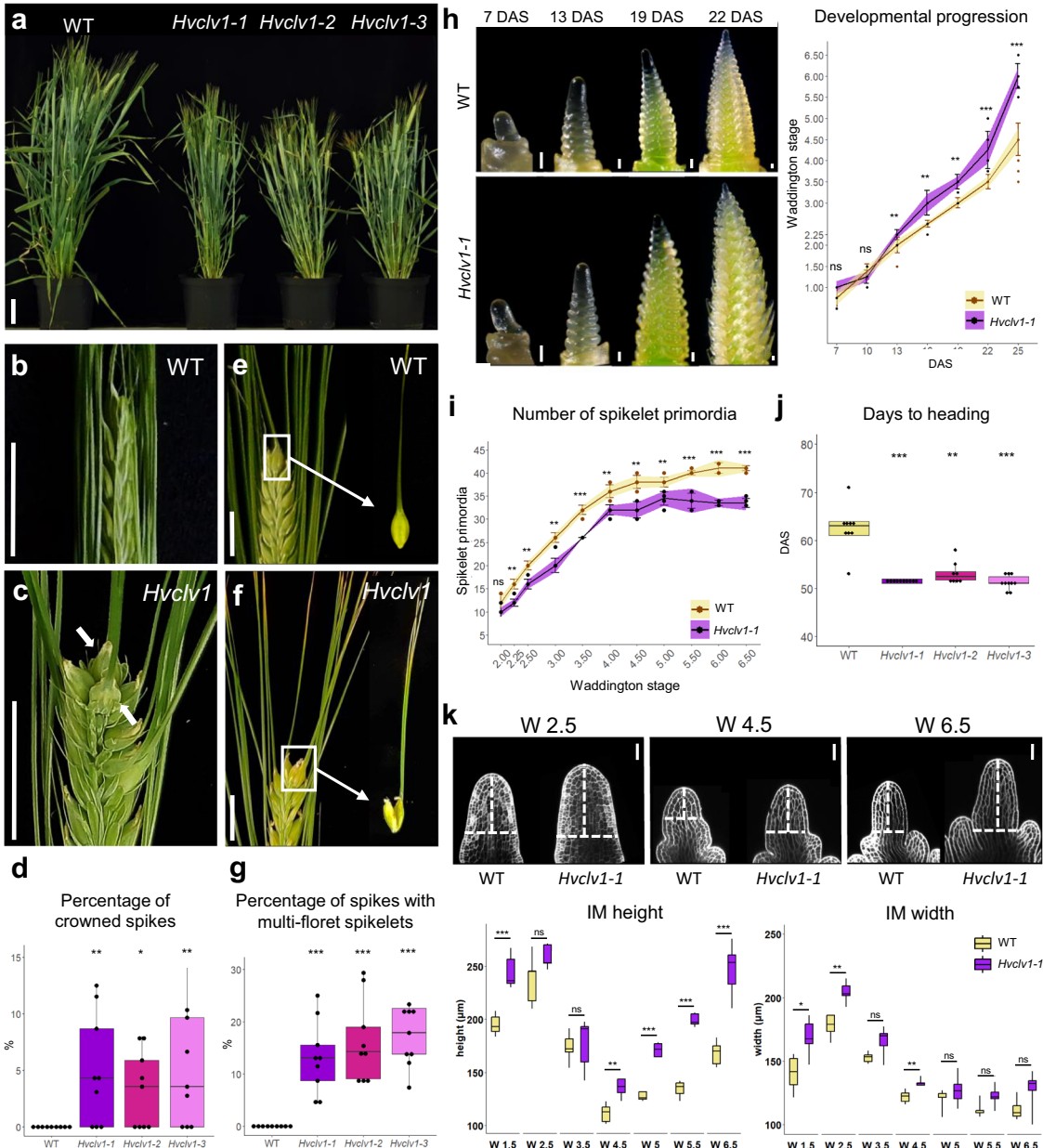

**Fig. 2 | HvCLV1 impacts plant and spike architecture, delays inflorescence development and promotes spikelet formation. a** *Hordeum vulgare cv.* Golden Promise Fast (WT) plants versus three selected *Hvclv1* mutant alleles (*Hvclv1-1*, *Hvclv1-2*, *Hvclv1-3*). **b, c** WT inflorescence and *Hvclv1-1* crowned spike phenotype, respectively; ectopic grains are indicated by white arrows. **d** Percentage of crowned spikes in WT and *Hvclv1* mutant alleles. **e, f** Close up on WT single grain and *Hvclv1-1* multi-grain developed from multi-floret spikelets. **g** Percentage of spikes with multi-grain in WT plants and *Hvclv1* mutant alleles. In (**d, g**) dots represent the percentage of spikes per plant exhibiting the phenotype from *n* = 9 plants over *n* = 3 independent experiments. **h, i** Stereo microscope pictures of WT and *Hvclv1-1* inflorescence development from 7 to 22 days after sowing (DAS) and quantification of developmental progression and number of spikelet primordia. WT in yellow, *Hvclv1-1* in purple. Dots: single measurements; error bars: standard deviation; coloured ribbon: interval of confidence. In (**h**) and (**i**) *n* = 10 and *n* = 7 inflorescences were analysed over *n* = 3 independent experiments, respectively. **j** Days required

for the main tiller to reach heading stage in WT and *Hvclv1* mutant alleles. *n* = 10 plants were sown over one experiment. Due to failure in grain germination, *n* = 9 and *n* = 8 data points were obtained from WT and *Hvclv1-2*, respectively. **k** Examples of WT and *Hvclv1-1* IMs at W2.5, 4.5 and 6.5. Stained cell wall in white. IM width and height at different W were measured by tracing a horizontal line from the last visible spikelet primordium (IM width) and a perpendicular vertical line connecting it to the highest IM point (IM height) in WT (yellow) and *Hvclv1-1* (purple) inflorescences (white segmented lines). *n* = 5 samples/genotype were measured for every W. Scale bar = 5 cm in (**a**), 1.5 cm (**b, c, e, f**), 100 μm (**h**) and 50 μm (**k**). Boxplots: median (centre line); upper and lower quartiles (box limits); 1.5x interquartile range (whiskers), dots (single measurements). Statistics: asterisks indicate the significant difference in comparison to WT using a two-sided Pairwise Wilcoxon rank sum test (**d, g, h, i, j**) or a two-sided Pairwise t-test (**k**). ns non-significant (*p*-value > 0.05); *(*p*-value < 0.05); **(*p*-value < 0.01); ***(*p*-value < 0.001).

that *Hvclv1-1* inflorescences develop similarly to WT along vegetative stages (7 − 10 days after sowing, DAS) (Fig. 2h), although *Hvclv1* vegetative SAMs (vSAMs) were slightly enlarged (see below). During the reproductive phase, *Hvclv1-1* inflorescences developed faster than WT, reaching each stage earlier, but also generated a reduced

number of spikelet primordia and underwent early termination of spikelet formation (Fig. 2h, i). The accelerated pace of spikelet development shown for *Hvclv1-1* resulted in early flowering. All three *Hvclv1* alleles reached the heading stage earlier than WT (Fig. 2j).

Detailed microscopic analysis of WT IM showed dynamic changes in its shape along development (Fig. 2k). At W1.5 (between 10 and 13 DAS), meristems start to produce spikelet primordia while cells still accumulate in the IM until W2.5. IM size is then reduced during rapid spikelet initiation (W2.5 to W4.5). After termination of spikelet formation at W5 to W5.5, the width of the IM remains unaltered, and cells start to accumulate along the vertical axis (Fig. 2i, k). IMs of *Hvclv1* were larger at W1.5 and always appeared more elongated than WT, except for the stages W2.5 and W3.5, when IM cells are consumed by rapid spikelet initiation (Fig. 2i, k). The length of the entire inflorescences was briefly reduced at early stages when *Hvclv1* mutants progressed rapidly through developmental stages but did not differ significantly from W4 onwards (Supplementary Fig. 2l). We conclude that *HvCLV1* first acts to restrict meristem growth and developmental progression of spikelet primordia, and promotes spikelet initiation at later stages. To track the origin of the multi-

floret spikelets and ectopic spikelet rows in *Hvclv1* plants, we imaged developing inflorescences by Scanning Electron Microscopy (SEM). Bases of *Hvclv1* IMs were enlarged at the initiation of spikelet formation (W1.5) (Fig. 2k), which correlated with the formation of an additional row of SMs in later stages. IMs then shifted from a distichous to a spiral phyllotaxis (Fig. 3a, b). In WT, the barley SM gives rise to the rachilla primordium (RP), which arrests development after initiation of a single floret (Fig. 3c–e). SEM analysis showed that the RP continues to grow wider and for an extended time in *Hvclv1* (Fig. 3f), forming either larger or additional florets, or even secondary RPs (Fig. 3g–j). Mature rachillae from *Hvclv1-1* were generally enlarged and exhibited a reduced formation of hair-like structures compared to the WT. *Hvclv1* rachillae occasionally formed a meristem-shaped structure at their tip, suggesting a proliferatively active state. (Supplementary Fig. 3c). We concluded that *HvCLV1* is required to restrict RP activities to the formation of a single floret.

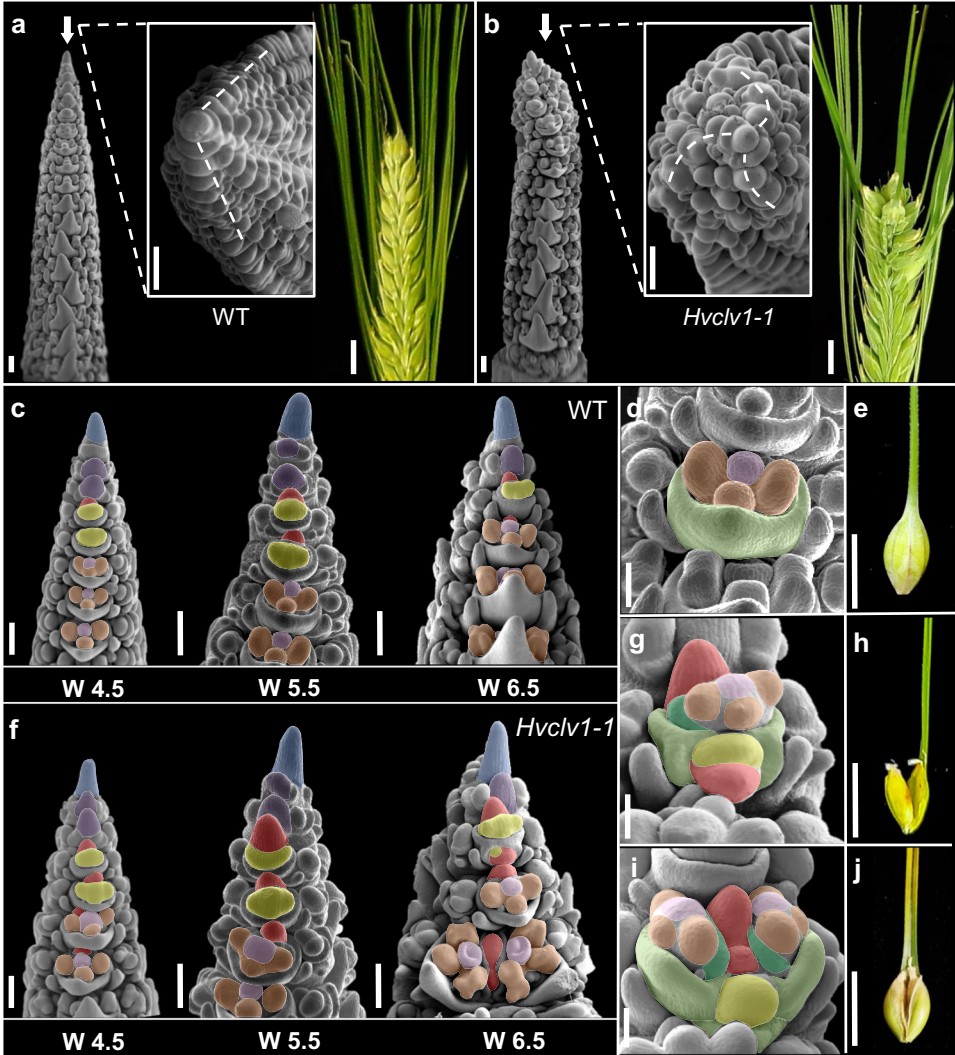

**Fig. 3 | The origin of crowned spikes and multi-floret spikelets. a, b** SEM pictures showing frontal and top view of WT and *Hvclv1-1* inflorescences at W4.5 respectively. Segmented lines indicate spikelet primordia phyllotaxis. Pictures of representative spike phenotypes on the right. Crowned spikes were found in 6/38 *Hvclv1-1* inflorescences from W5.5 to W6.5 **c** Developmental progression of WT inflorescences at W4.5, 5.5 and 6.5. Colour code as described (see above). **d, e** Close-up on WT floret and WT single grain respectively. **f** Developmental progression of *Hvclv1-1* inflorescences at W4.5, 5.5 and 6.5. Colour code as described (see above). Multi-floret spikelets were found in 27/38 *Hvclv1-1* inflorescences from W5.5 to W6.5. **g–j** Close up on *Hvclv1-1* multi-floret spikelet and the resulting multi-grain disposed vertically (**g, h**) and horizontally (**i, j**). Multi-floret spikelets disposed vertically were observed in 24/38 inflorescences and horizontally in 10/38 inflorescences. All the inflorescences for SEM pictures were collected from the main tiller of plants grown under the same conditions over *n* = 4 independent experiments. Colour code (**g–j**): RP and secondary rachilla in red, FM in yellow, LP in light green, PP in dark green, SP in brown and CP in pink. Scale bars: **a–c**, **f** = 200 μm; **d, g, i** = 100 μm; **e, h, j** = 1 cm.

### The CLE peptide HvFCP1 acts with HvCLV1 to limit meristem activities

In many plant species, CLV1-family receptors were found to interact with CLE peptides. In barley, we identified a total of 28 CLE peptides, including an ortholog of CLV3 (HvCLV3) and an evolutionarily conserved FCP1-like peptide, which we named HvFCP1 (Supplementary Fig. 5a–c). Incubation of WT barley seedlings with growth medium containing 30 μM of synthetic HvFCP1 peptide caused a reduction in meristem height of the WT vSAM, while vSAM width was not affected. *Hvclv1-1* mutant seedlings were insensitive to HvFCP1 treatment (Fig. 4a), indicating that HvFCP1 requires HvCLV1 to limit vSAM height. We then analysed the expression of a transcriptional reporter line, *pHvFCP1:mVenus-H2B*, in transgenic barley (Fig. 4b–e). During the vegetative phase, the *HvFCP1* promoter was active in the vSAM, but not in leaf initiation sites (Fig. 4b). Later on, activity was found in the IM and from the triple spikelet meristem stage onwards (Fig. 4c). Moreover, the activity was polarised at the adaxial side of the developing central spikelet and later in the fully formed RP (Fig. 4e). In FMs, *HvFCP1* was mainly expressed in the central domain and later in carpel primordia (Fig. 4d). Importantly, the *HvFCP1* reporter was more prominently expressed in the RP compared to *HvCLV1* (Supplementary Fig. 6A–J).

We generated two independent knock-out mutant alleles by CRISPR-Cas9 (*Hvfcp1-1* and *−2*) to study HvFCP1 function. Both *Hvfcp1* alleles are phenotypically indistinguishable, and molecular analysis identifies them as loss-of-function mutants (Supplementary Fig. 7a, Supplementary Notes 2). *Hvfcp1* plants, similar to *Hvclv1*, remained shorter with shorter inflorescences and formed fewer viable grains, while tiller or internode number was not affected (Supplementary Fig. 7b–e). *Hvfcp1* mutant IM height and width increased similarly to those of *Hvclv1* mutants and developed faster than WT (Supplementary Fig. 7f–h). Ectopic formation of spikelet rows was not observed, but we found multi-floret spikelets as described for *Hvclv1* (Fig. 4f, g; Supplementary Fig. 7i), indicating that HvFCP1 acts with HvCLV1 in the regulation of vSAM, IM, SM and RM determinacy. We next analysed the roles of *HvCLV1* and *HvFCP1* in regulating meristem growth using confocal imaging and cell segmentation followed by computational 3D reconstructions of WT, *Hvclv1-1* and *Hvfcp1-1* IMs from W4.5 to W6.5 (Fig. 4h). After W5 and termination of spikelet formation, the sizes of *Hvclv1* and *Hvfcp1* IMs increased more rapidly than WT, due to enhanced cell proliferation (Fig. 4i,j; Supplementary Fig. 8a–d). Sizes of the RP were also increased at all stages in *Hvclv1-1* and *Hvfcp1-1*, compared to WT (Fig. 5a, b), while FMs were less affected (Fig. 5c). *Hvfcp1* mutant phenotypes were overall milder than those of *Hvclv1*, and loss of *HvFCP1* activity did not enhance the phenotype of *Hvclv1* in the *Hvclv1;Hvfcp1* double mutant (Fig. 5d), indicating that other CLE peptides can partially compensate for the loss of HvFCP1 and signal through HvCLV1. To analyse this further, we crossed the *HvCLV1* and *HvFCP1* reporter lines into the *Hvfcp1* and *Hvclv1* mutant backgrounds (Fig. 5e–l). Expression of *pHvCLV1:HvCLV1-mVenus* was unaltered in the *Hvfcp1-1* IM (Fig. 5e, f), while *HvFCP1* expression was reduced in *Hvclv1-1* (Fig. 5i–l), indicating that HvFCP1 is subject to positive feedback regulation. Interestingly, HvCLV1 protein internalization, which indicates receptor turnover upon ligand binding[31], was significantly reduced in the RP of *Hvfcp1*, but still detected, suggesting that in the absence of HvFCP1, another CLE peptide can partially compensate its function through HvCLV1 (Fig. 5g, h; Supplementary Fig. 8e).

### HvFCP1 and HvCLV1 control meristematic proliferation through coordination of cell division, auxin signalling and trehalose-6-phosphate

To investigate the common function of HvCLV1 and HvFCP1, we performed RNA-sequencing of WT, *Hvclv1-1* and *Hvfcp1-1* inflorescences at W3.5. A total of 1,208 genes were upregulated and 1,197 downregulated in *Hvclv1* vs WT, while 521 and 258 were upregulated and downregulated in *Hvfcp1* vs WT, respectively. Interestingly, 55.2% (288) of the upregulated and 39.9% (103) of the downregulated genes in *Hvfcp1* vs WT were in common with *Hvclv1* vs WT, suggesting a significantly shared function of HvFCP1 in the larger gene regulatory network affected by HvCLV1 (Supplementary Fig. 9a, b, Supplementary Data 2).

Mutation of the HvFCP1/HvCLV1 signalling pathway resulted in enhanced proliferation of the IM and RP compared to WT, which ultimately repressed spikelet formation and promoted inflorescence branching. Within the similarly upregulated genes in *Hvclv1* vs WT and *Hvfcp1* vs WT, we found *HvBG1*, an ortholog of *Rice Big Grain1* (RBG1), which promoted cell division and auxin accumulation in meristematic and proliferating tissues when overexpressed in rice[32]. Furthermore, upregulation of the P-type cyclin *HvCYCP4-1* and the bicistronic transcript encoding Triphosphate Tunnel Metalloenzyme 3 (HvTTM3) and CELL DIVISION CYCLE PROTEIN26 (HvCDC26), together with upregulation of the auxin response genes *HvIAA13* and *HvIAA31*, indicated a general promotion of cell division and alteration of auxin signalling[33–35]. Interestingly, smRNA-FISH results revealed an increased number of *HvIAA31* transcripts in the RP and main rachis of *Hvclv1* inflorescences compared to WT (Supplementary Fig. 10a). Additionally, HvBAM5, an LRR-RLK closely related to HvCLV1, was upregulated in *Hvclv1* vs WT, possibly compensating for aspects of the mutant phenotype. Several genes known to be involved in spikelet development were differentially expressed in *Hvclv1* and *Hvfcp1* compared to WT. HvMND6 was upregulated in both mutants, which encodes a Cytochrome 450 enzyme, whose loss of function was previously associated with increased leaf production and reduced plant stature[36]. In WT inflorescences at W3.5, HvMND6 expression was mostly restricted to the spikelet boundaries and suppressed bracts, whereas, in *Hvclv1* inflorescences, its expression expanded to the main rachis and RP (Supplementary Fig. 10b).

The MADS-box transcription factors HvMADS1 and HvMADS3 were significantly downregulated in *Hvclv1* vs WT. In WT inflorescences, *HvMADS1* was expressed in the RP and LEP, while *HvMADS3* expression specifically marked the floral organs. However, smRNA-FISH results did not show a notable reduction in *HvMADS1* and *HvMADS3* transcript levels in their respective expressing tissues, suggesting that their general downregulation was likely correlated with the reduced number of spikelet and floret primordia in *Hvclv1* inflorescences compared to WT. Interestingly, *HvMADS1* was ectopically expressed in the IM of *Hvclv1*, implying a possible function of HvCLV1 in restricting its expression to the spikelet organs (Supplementary Fig. 10c, d). Additionally, the barley gene *COM2* was downregulated in Hv*clv1* vs WT, suggesting a role of HvCLV1 in the upstream regulation of this transcription factor involved in the repression of spike branching.

Inflorescence branching was previously associated with increased levels of Threalose-6-Phospate (T6P). Mutation of the maize gene *RAMOSA3* (*RA3*), encoding a Threalose-6-Phospate Phosphatase, led to indeterminate growth of inflorescence auxiliary meristems, that produced long branches bearing additional FMs[16]. Moreover, studies in Arabidopsis linked increased levels of T6P in axillary meristems with enhanced shoot branching via *FLOWERING LOCUS T* (*FT*) and upregulation of the sucrose transporter *Sugars Will Eventually be Exported Transporters11* (*SWEET11*)[37].

In both *Hvclv1* vs WT and *Hvfcp1* vs WT, *SISTER OF RAMOSA3* (*HvSRA*), paralogue of the maize *RA3*, was downregulated, and *HvTPS1*, the closest ortholog of the Arabidopsis Threalose-6-Phosphate Synthase1 (*TPS1*) was upregulated, suggesting an impaired T6P metabolism. Consistent with findings in Arabidopsis, the sucrose transporter *HvSWEET11b* and *HvFT2*, a barley paralogue of FT, were upregulated in both mutants in comparison to WT, indicating a general reallocation of sucrose and alteration of SM identity (Fig. 6a, Supplementary Fig. 9c)[38,39].

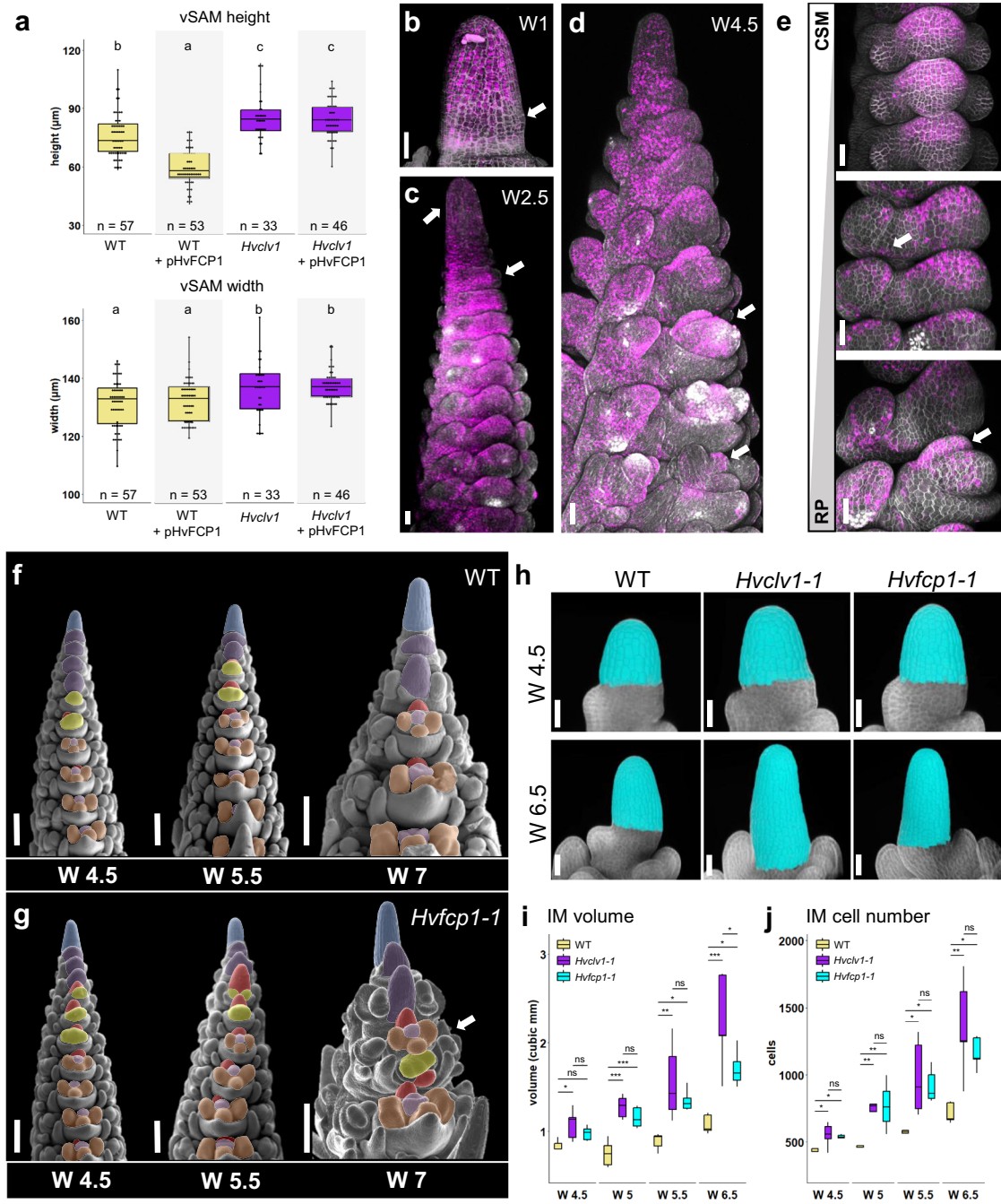

**Fig. 4 | HvFCP1 interacts with HvCLV1 to regulate IM and RP size and determinacy. a** Vegetative shoot apical meristem (vSAM) height and width in control samples (white background) and samples treated with HvFCP1 synthetic peptide (pHvFCP1) (grey background). WT vSAMs in yellow and *Hvclv1-1* in purple. Dots represent single measurements, *n* number of samples over *n* = 3 independent experiments. The letters on top of each boxplot represent the results of a two-sided Pairwise Wilcoxon rank sum test. **b**–**d** Confocal images of barley SAM and inflorescence expressing *HvFCP1* transcriptional reporter line (pHvFCP1:mVenus-H2B) at different developmental stages and in different organ primordia. SAM at vegetative stage (**b**), at W2.5 (**c**) and W3.5 (**d**). **e** *HvFCP1* transcriptional reporter line along rachilla development. From central spikelet meristem (CSM) to rachilla primordium (RP). Inflorescence phenotype in late stages of development (W4.5, W5.5, W7) in WT

(**f**) and *Hvfcp1-1* (**g**). Multi-floret spikelets were found in 12/35 *Hvfcp1-1* inflorescences from W5.5 to W6.5 over *n* = 4 independent experiments. Colour code as described (see above) **h**–**j** 3D reconstruction of WT, *Hvclv1-1* and *Hvfcp1-1* IMs at W4.5 and W6.5 (**h**). Cells in cyan were selected for the IM measurements and *n* = 5 IMs were analysed for each genotype and developmental stage over *n* = 3 independent experiments in (**i**) and (**j**). A horizontal line was drawn from the last visible primordium, and all the above cells were considered part of the IM. Boxplots display IM volume and cell number, respectively. WT (yellow), *Hvclv1-1* (purple), *Hvfcp1-1* (cyan). Asterisks indicate the significant difference to WT using a two-sided Pairwise t-test. Boxplots: median (centre line); upper and lower quartiles (box limits); 1.5x interquartile range (whiskers). Scale bars: 50 µm (**b**–**e**, **h**) and 200 µm (**f**, **g**). Statistics: ns non-significant ($p$-value > 0.05); *($p$-value < 0.05); **($p$-value < 0.01); ***($p$-value < 0.001).

## Discussion

In this study, we characterised the function of *CLAVATA* signalling components in coordinating the activity of different meristems within the barley inflorescence and showed that *HvCLV1* with *HvFCP1*

regulates IM and RP proliferation and determinacy. The localised expression of *HvFCP1* overlapped with only part of the broader expression of *HvCLV1* (Fig. 6b), and while both *Hvclv1* and *Hvfcp1* mutants developed multi-floret spikelets as a consequence of their

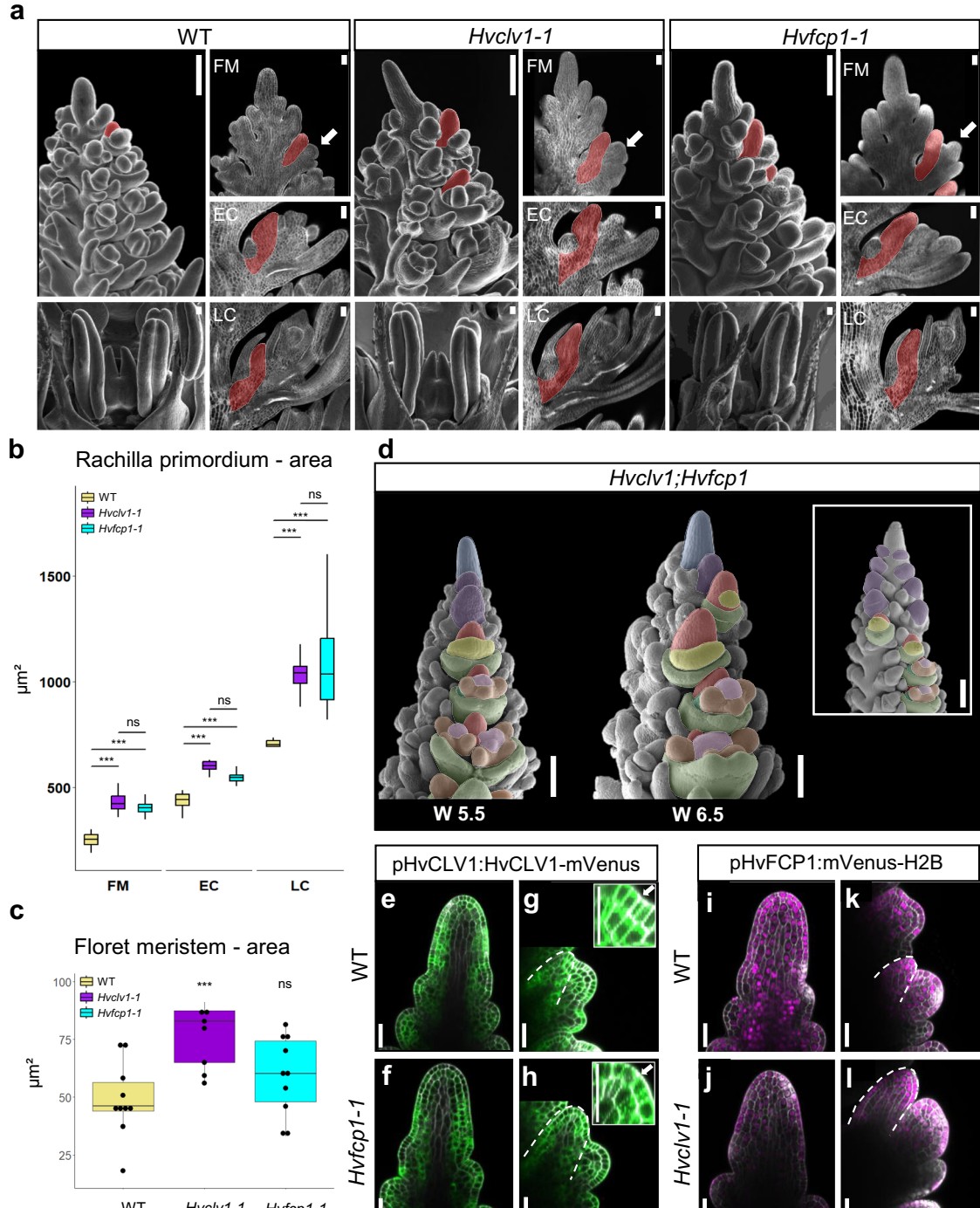

**Fig. 5 | HvCLV1 and HvFCP1 repress RP elongation, *Hvclv1;Hvfcp1* double mutant and reporter lines in the respective mutant backgrounds. a** SEM pictures of WT, *Hvclv1-1* and *Hvfcp1-1* inflorescences at W6.5 were used as a reference for matching longitudinal sections on their right. The RP is highlighted in red in the three stages of flower development: floret meristem (FM), early carpel (EC), and late carpel stage (LC). White arrows indicate the spikelet stage considered as FM within the image. **b** Boxplots displaying rachilla area from central longitudinal sections in WT (yellow), *Hvclv1-1* (purple), *Hvfcp1-1* (cyan). Asterisks indicate the significant difference to WT using a two-sided Pairwise Wilcoxon rank sum test. *n* = 8 inflorescences were analysed for each genotype and used to image spikelets at different developmental stages over *n* = 2 independent experiments. **c** Measurements of FM area from SEM frontal pictures of barley inflorescences at W5.5 in WT, *Hvclv1-1* and *Hvfcp1-1*. Dots represent biological replicates, and asterisks indicate the significant

difference to WT using a two-sided t-test. *n* = 10 SEM images over *n* = 4 independent experiments. **d** Inflorescence phenotype of *Hvclv1;Hvfcp1* at W5.5 and W6.5, which exhibited occasional crowned spike phenotype. Colour code as described (see above). HvCLV1 proteins localisation (green) in WT (**e**, **g**) and *Hvfcp1-1* (**f**, **h**) IM and RM (segmented line) respectively. **i**–**l** *HvFCP1* expression pattern (magenta) respectively in WT and *Hvclv1-1* IM. (**r**, **s**) *HvFCP1* expression pattern (magenta) in WT (**i**, **k**) and *Hvfcp1-1* (**j**, **l**) IM and RM (segmented line), respectively. Boxplots: median (centre line); upper and lower quartiles (box limits); 1.5x interquartile range (whiskers); points, outliers. Scale bars in **a**: SEM pictures of the inflorescence tip = 200 μm; SEM pictures of flowers and all sections = 50 μm, in **d** = 200 μm, in **e**–**l** = 50 μm. Statistics: ns non-significant (*p*-value > 0.05); *(*p*-value < 0.05); **(*p*-value < 0.01); ***(*p*-value < 0.001).

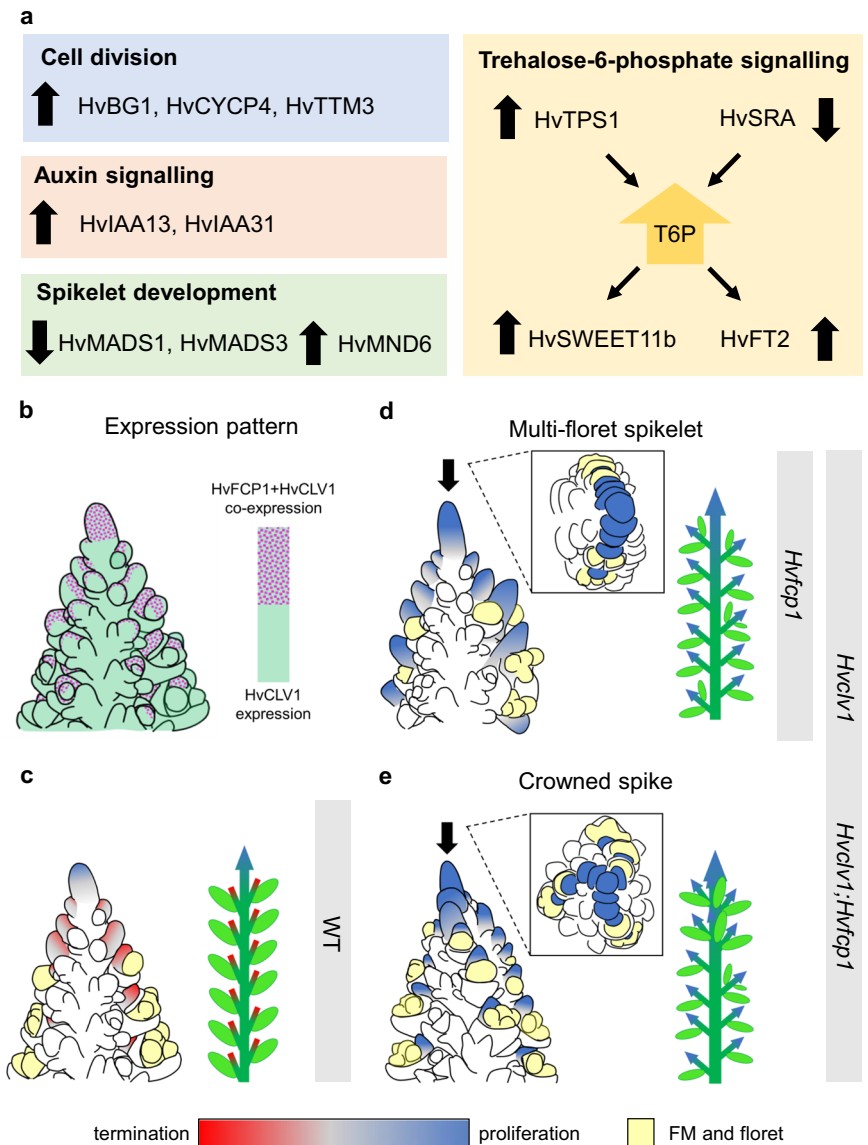

**Fig. 6 | Comparative transcriptome analysis of similarly regulated genes in *Hvclv1* and *Hvfcp1* vs WT, and schematic summary. a** Similarly regulated genes in *Hvclv1* vs WT and *Hvfcp1* vs WT from RNA sequencing results. Black arrows pointing upward indicate upregulated genes, while arrows pointing downward indicate downregulated genes. **b** Schematic representation of HvCLV1 (green) and *HvFCP1* (magenta dots) expression patterns in barley inflorescence at W5.5. **c**–**e** Schematic representation of barley inflorescences at W5.5 (left) and mature spikes (right) in WT (**c**), *Hvfcp1*, *Hvclv1* and *Hvclv1;Hvfcp1* (**d**, **e**). Grey bars indicate the observed spike phenotypes (multi-floret spikelets and crowned spikes) in the respective genetic backgrounds. All elements in (**b**–**e**) were drawn using PowerPoint. Colour code: IM and RMs are marked in red or blue to indicate meristematic proliferation or termination, respectively. FM and floral organs are marked in yellow; the main rachis and grains are marked in dark and light green.

indeterminate and enlarged rachilla, only *Hvclv1* develop crowned spikes. Both mutants' RP produced additional florets, implying a change in its determinacy, and partially reproducing the behaviour of the indeterminate rachilla of wheat[2,10]. Additionally, the overall weaker phenotype of *Hvfcp1*, together with observation of HvCLV1 protein internalisation in *Hvfcp1* background and the *Hvclv1;Hvfcp1* double mutant phenotype, suggested that other, not yet identified CLE peptides could interact with HvCLV1 and partially rescue the *Hvfcp1* mutant phenotype (Fig. 6c–e). Altogether, our results point towards a more general role of HvCLV1 in mediating the downstream transmission of signals triggered by specifically expressed CLE peptides in order to regulate the proliferation of different meristems. The increased size of the IM in *Hvclv1* and *Hvfcp1* caused a reduced formation of spikelet and grains, corroborating the negative correlation between IM size and grain yield previously observed in barley

FASCIATED EAR (HvFEA) mutants[40]. Transcriptional analyses of *Hvclv1* and *Hvfcp1* apices highlighted a shared regulatory network between HvCLV1 and HvFCP1, directly or indirectly controlling the expression of genes involved in cell division, auxin signalling, and T6P metabolism. Upregulation in the main rachis and RP of the auxin response factor HvIAA31 and the cytochrome P450 gene *HvMND6* in *Hvclv1* compared to WT not only confirmed the RNA-seq results but also suggested a spatially restricted alteration of auxin signalling and metabolic pathways in rachis-like structures. An enhanced activity of the RP was accompanied by upregulation of *HvTPS1* and down-regulation of *HvSRA*, which likely results in the accumulation of T6P, previously linked with enhanced branching in Arabidopsis, pea and maize[16,37,41]. Interestingly, HvTPS1 and HvSRA were differentially regulated in branched barley mutants, such as *vrs4* and *com1*[11,13]. However, both VRS4 and COM1 were not differentially regulated in *Hvclv1* or

*Hvfcp1* vs WT, suggesting an independent regulation of HvTPS1 and HvSRA by CLV signalling, possibly through a pathway including COM2, which was downregulated in *Hvclv1* vs WT. Increased T6P levels were shown to lead to the reorganisation of sugar transport by transcriptional regulation of *SWEET* genes and upregulation of *FT*-related genes, which are directly involved in spikelet identity and flowering time[37]. *HvFT2* overexpression in barley consistently resulted in early flowering plants with reduced formation of spikelet primordia, similar to the phenotype observed in *Hvclv1* and *Hvfcp1*[42].

Meristem homoeostasis was demonstrated to be controlled redundantly by different receptors and peptides. In maize, rice, and Arabidopsis, parallel and antagonistic pathways were shown to control IM shape and maintenance[21,43,44]. The relatively mild phenotype of *Hvclv1*, in comparison to the phenotypes described for *td1* in maize or *clv1* in Arabidopsis, is probably the result of partial compensation by additional CLV-related receptors acting in parallel, such as HvBAM5, which was upregulated in *Hvclv1* vs WT[26,45].

Our study showed how the *CLV* signalling pathway orchestrates the determinacy and growth of diverse meristems comprising the barley spike by coordinating meristematic cell division and auxin signalling with levels of T6P, which acts as a signalling metabolite linking meristem growth with sugar allocation and flowering time. Here, we notice an underexplored opportunity to redesign and optimise barley inflorescence architecture via specific regulation of distinct meristem activities. The large diversity of inflorescence architectures that already evolved in grasses indicates that the underlying genetic networks offer a vast, yet hidden potential to encode a much wider morpho-space than what is realised in our current cereal varieties.

# Methods

## Phylogenetic analysis

The CLV1 clade was identified by phylogenetic reconstruction of the protein kinase superfamily of four monocots (*Oryza sativa, Triticum turgidum, Zea mays*, and *Hordeum vulgare*), and two dicot species (*Arabidopsis thaliana, Solanum lycopersicum*). The proteomes of these species were downloaded from EnsemblePlants (https://plants.ensembl.org/index.html). To identify all protein kinase domain containing proteins in the selected species' proteomes, we conducted HMMscan using HMMER V 3.2 (http://hmmer.org)[46]. HMM profiles of the protein kinase domain (Pfam 10.0) were downloaded from InterPro[47] to carry out the HMM matching. Based on the HMMscan result an E-value threshold of <1e−10 was imposed to identify the protein kinase domains (PF00069) in the given protein sequences. Protein kinase domains were extracted from all the protein sequences using custom Python scripts (https://github.com/Thilanka-lt/Pfam_domain_tree_construction) and were subjected to HMMalign for protein kinase domain alignment[46]. The multiple sequence alignment was used to construct a Neighbour-joining tree (using JTT + CAT model and default parameters) using the FastTree package[48]. Based on the constructed protein kinase domain family tree we identified the CLV1 clade. Next, we extracted the protein kinase domain sequences from the operational taxonomic units of the selected CLV1 clade to further examine the phylogenetic relationships of kinase domains within the CLV1 clade. We constructed the receptor-like kinase phylogeny using RAxML (using random seed for tree initiation and non-parametric bootstrapping) for 1000 bootstrap replicates[49]. This was completed using automated model selection criteria to select the best evolutionary model that fits the dataset.

## Plant material and growth conditions

All barley plants used in this study were *cv*. Golden Promise Fast[50] and were grown in soil (Einheitserde ED73, Einheitserde Werkverband e.V., with 7% sand and 4 g/L Osmocote Exact Hi.End 3–4 M, 4th generation, ICL Group Ltd.) under long day (LD) conditions with 16 h light at 20 °C and 8 h dark at 16 °C. Plants used for microscopy were grown in

QuickPot 96 T trays (HerkuPlast Kubern GmbH) in a climate chamber, while the plant phenotype was described in plants growing in larger pots (diameter 16.5 cm, height 13 cm) in a greenhouse under the same growing conditions but with temperatures that slightly varied between seasons. Grains were always pregerminated in Petri dishes with water at 4 °C for 3 days before being sowed in soil.

## Quantification of HvCLV1 internalization in the cytoplasm

To estimate differences in HvCLV1 internalisation in the RP and FM we made use of high-resolution confocal images of spikelets expressing pHvCLV1:HvCLV1-mVenus and treated with DAPI (see below) (Supplementary Fig. 1C). Pictures were analysed using the software Fiji. The YFP channel showing the localisation of HvCLV1 proteins was separated from the DAPI channel staining the cell wall. The maximum fluorescent intensity of each channel was normalised to a value of one. For each spikelet section, two ROIs delimiting DAPI-stained cells of the RP and FM were drawn. The DAPI signal was subtracted from the YFP channel, erasing the mVenus signal from the plasma membrane. The remaining cytoplasmatic mVenus signal was transformed into a mask using a default percentile threshold of 0.5 (Supplementary Fig. 1D). In each pixel of the generated mask the fluorescence signal was translated into a grey value from 0 (black) to 255 (white) which allows the quantification of its brightness. The average grey value of the RP or FM was calculated as the sum of the grey values of each pixel divided by the total number of pixels included in the selected ROI.

## Plasmid construction and plant transformation

The *pHvCLV1:HvCLV1-mVenus* plasmid was constructed by PCR amplification of a 2826 bp fragment upstream of the start codon of *HvCLV1* (HORVU.MOREX.r3.7HG0747230) as putative regulatory sequence from Morex genomic DNA (gDNA) and cloned by restriction and ligation via a AscI site into a modified pMDC99[51]. The *HvCLV1* coding region without stop codon (3573 bp) was amplified from Morex gDNA and inserted downstream of the promoter by Gateway cloning (Invitrogen). A C-terminal mVENUS was integrated downstream of the gateway site by restriction and ligation via PacI and SpeI (Supplementary Table 1). The *pHvFCP1:VENUS-H2B* construct was cloned by amplifying the regulatory sequence including 2034 bp upstream of the start codon of *HvFCP1* (HORVU.MOREX.r3.2HG0174890) and inserted by Gateway cloning (Invitrogen) into the modified pMDC99[51]. This modified pMDC99 contained the gateway cassette, the coding sequence of VENUS and a T3A terminator, which were inserted by restriction via AscI and SacI from pAB114[52]. Furthermore, it contains the coding sequence of Arabidopsis HISTONE H2B (AT5G22880) at the C terminus of VENUS for nuclear localization, inserted via restriction and ligation at a PacI restriction site (Supplementary Table 1). Both *pHvCLV1:HvCLV1-mVenus* and *pHvFCP1:VENUS-H2B* constructs were first transformed in the barley cultivar Golden Promise[53] and then crossed into Golden Promise Fast. *Hvclv1* and *Hvfcp1* mutant alleles were generated by CRISPR-Cas9 genome editing. Plasmids were constructed using the vector system and following the established protocol[54]. The *HvCLV1* gene was targeted by a single 20 bp sgRNA 53 bp after the coding sequence started, while two 20 bp sgRNAs were cloned to target *HvFCP1* 298 and 542 bp after the start codon. All sgRNAs were designed using E-CRISP software[55] and single sgRNA strands were hybridized and cloned into the shuttle vectors pMGE625 or pMGE627 by a BpiI cut/ligation reaction. A second cut/ligation reaction (BsaI) was used to transfer the gRNA transformation units (TUs) to the recipient vector pMGE599[54]. The final vector targeting *HvCLV1* was transformed in Golden promise Fast via embryo transformation[56], while the vector targeting *HvFCP1* was transformed in Golden Promise Fast via embryo transformation, but using the transformation protocol by Hensel et al.[57]. Successful insertion of the transformation vector into the genome was tested by PCR (Supplementary Table 1) on M0 plants. The Cas9 protein was removed by

segregation in M1 plants, and homozygous mutations of *HvCLV1* and *HvFCP1* were identified in M2 plants by amplification of genomic sequences targeted by the sgRNAs and subsequent Sanger sequencing. (Supplementary Table 1).

## smRNAfish

Barley inflorescences at W3.5 fixed in 4% PFA were embedded in paraplast (Leica Paraplast X-tra) and tissue sections (10 μm) were placed within the capture areas on Resolve Bioscience slides and incubated on a hot plate for at least 20 min at 60 °C to attach the samples to the slides. Slides were deparaffinised, permeabilised, acetylated and re-fixed. Sections were mounted with a few drops of SlowFade-Gold Antifade reagent (Invitrogen) and covered with a coverslip to prevent damage during shipment to Resolve BioSciences (Germany).

## Plant and grain phenotyping

WT, *Hvclv1* and *Hvfcp1* plants phenotyping was performed at the end of their life cycle when completely dried. Plant measurements and percentage of crowned spikes and multi-grains were performed in all the tillers with no distinction between main stem and lateral branches, since the spike phenotypes raised with the same probability in both main and lateral tillers. Three replicates were performed and three plants per replicate were phenotyped. Grain measurements were obtained using MARViN (MARViTECH GmbH).

## Sample preparation, microscopy and image processing

Barley SAMs and inflorescences were collected by manual removal of all the surrounding leaves. Smaller leaves were dissected under a stereo microscope using a 1.5 mm blade scalpel. Fresh inflorescences were directly imaged for stereo microscope pictures using a Nikon SMZ25 stereo microscope with Nikon DS-Fi2 camera. For confocal imaging, fresh barley inflorescences were stuck on their side on a double-sided adhesive tape on an objective slide, stained with 4′,6-diamidin-2-phenylindol (DAPI 1 μg/mL) for 3 min, washed three times with water and subsequently covered with a cover slide before being placed under the microscope. Confocal imaging was performed using Zeiss LSM780 and Zeiss LSM880 with a EC PlnN 10×/0.3, Plan-Apochromat 20×/0.8 or Plan-Apochromat 40x/1 objectives. SEM pictures were obtained by direct imaging of fresh inflorescences or by imaging epoxy replicas of barley inflorescences. At first a negative imprint of the inflorescence was created by mixing the two-component vinyl polysiloxane impression material (Express™ 2Ultra Light Body Quick, 3 M ESPE) and pushing the dissected inflorescence into the impression material, which polymerizes a few minutes after having been mixed. After complete polymerization of the negative print, the plant material was removed, and the negative print was filled with epoxy resin. After over-night polymerization, inflorescence replicates were coated with gold using an Agar Sputter Coater and imaged with a Zeiss SUPRA 55VP SEM.

## Peptide treatment

Barley WT and *Hvclv1-1* embryos were dissected at 10 days after pollination, when the SAM was exposed, and cultured on gel media. The medium was prepared by mixing 4.4 g/L of MS medium, 2% sucrose, and 500 ul/L of iron chelate. The pH was adjusted to 6.0 before the addition of 1.5 g/L Gelrite. The medium was then autoclaved. Before being poured in 12 cm square plates, after the medium cooled down, 1/1000 v/v ratio of vitamin mix was added. Treated embryos were then grown in medium with HvFCP1 synthetic peptide (REVPTGPDPIHH, by peptides&elephants GmbH) dissolved in 50 μl DMSO, reaching a final peptide concentration of 30 μM, while control embryos were grown in medium with 50 μl DMSO. Plates with embryos were grown in a Phyto-cabinet for 30 days at 24 °C under long day conditions. After 30 days, when young seedlings developed, vegetative meristems were dissected, fixed in 4% PFA overnight, washed three times with 1X PBS and

incubated for one week in ClearSee solution[58]. Pictures of the cleared meristems were taken under the Zeiss Axioskop 2 light microscope with AxioCam HRc camera. Images were then analysed in Fiji. SAM width was measured by the length of a horizontal line drawn across the SAM base, just on top of the last visible leaf primordium. Meristem height was calculated as the distance between the SAM tip and the centre of the horizontal line defining the base of the SAM. Three replicates of this experiment were performed, and forty embryos were plated for each replicate, even though not all the embryos germinated.

## IM 3D reconstruction and rachilla vibratome sections

IMs at different W stages were fixed in 4% PFA overnight, then washed three times with water and cleared in ClearSee solution for at least two weeks[58]. One day before imaging, 1/1000 v/v of SR 2200 stain by Renaissance Chemicals was added to the ClearSee solution and the cell wall was stained. After three washing steps in 1X PBS, barley inflorescences at different stages were glued on the bottom of a small petri dish with a drop of super glue and covered in 1X PBS. The petri dish was placed under Zeiss LSM900 confocal microscope and a z-stack of the submerged IM was imaged from the top with a 20×/0.5 water dipping objective. The 3D reconstruction was performed by loading the IM z-stacks in MorphoGraphX 2.0 and analysed accordingly to the protocol[59]. Rachilla central longitudinal sections were obtained by following the same procedure described above for the 3D reconstruction. The fixed, cleared inflorescences at W6.5 were embedded in 6% agarose in Disposable Base Molds (epredia) and 50 μm sections were obtained using a Leica VT1000S vibratome. The sections were stained in a Petri dish in 1X PBS with 1/100 v/v concentration of SR 2200 stain for a few minutes, placed on an objective slide with 1X PBS, and covered with a cover slide. Sections of 10 inflorescences for each genotype were then imaged under a Zeiss LSM880 confocal microscope.

## RNA-sequencing

To detect gene expression changes in *Hvclv1* and *Hvfcp1* inflorescence in comparison to WT, we collected inflorescences of WT, *Hvclv1-1*, and *Hvfcp1-1* at W3.5 for RNA-sequencing. Each replicate contained 40 pooled inflorescences from the main shoot of individual plants. All samples were collected manually under a stereo microscope without surrounding leaves. A total of three biological replicates of each genotype were used for RNA sequencing. Total RNA was extracted from inflorescences using the Direct-zol™ RNA, Miniprep Plus following the manufacturer's instructions and digested with DNase I (ZYMO RESEARCH). RNA samples passing a cutoff of RNA Integrity Number (RIN) ≥ 8 were used for mRNA library preparation using the poly-A enrichment method. Sequencing was performed on Illumina Novaseq 6000 sequencing platform (PE150), and at least 6 G of clean reads data per sample were generated by Biomarker Technologies (BMK) GmbH. To quantify transcripts, all clean reads were mapped to the Morex reference Version 3[60] using Salmon (v. 0.14.1)[61]. We kept transcripts with a minimum of 1 CPM (counts per million) in at least three samples. Analyses were conducted on 22,307 expressed genes. To identify differentially expressed genes (DEGs) within *Hvclv1* vs WT and *Hvfcp1* vs WT, a pairwise comparison was conducted using the count-based Fisher's Exact Test in R package 'EdgeR' (v3.32.1)[62]. The FDR of each gene was adjusted by the Benjamini-Hochberg (BH) procedure, thus genes with BH.FDR < 0.05 and $\log_2 FC \leq -0.5$ or $\log_2 FC \geq 0.5$ were referred to as downregulated or upregulated, respectively. The heatmap of gene expression (Supplementary Fig. 9C) was generated on all the differently expressed genes in *Hvclv1* vs WT and *Hvfcp1* vs WT with $-\log_{10}(TPM + 1)$ values using 'ThreeDRNAseq' R package[63].

## Quantification and statistical analysis

All the statistical tests were performed using R Studio (RStudio Team 2022). For each data set we performed Shapiro–Wilk normality test (function shapiro.test from package stats, v3.6.2) to verify its normal

distribution, and Levene's Test (function lavene.test from package stats, v3.6.2) to verify the homogeneity of its variance. When two groups were normally distributed and with a homogeneous variance (Shapiro–Wilk test and Levene's Test showing $p$-values ≥ 0.05), a 2-tailed, unpaired Student's $t$ test (function t_test from the package rstatix, v0.7.2) was used to determine their significant difference, with a P-value cutoff at ≤0.05. When two groups were not normally distributed and/or not displaying a homogeneous variance (Shapiro–Wilk test and Levene's Test showing $p$-values < 0.05), an unpaired Post-hoc's Wilcoxon test was performed. When testing significant differences between more than two groups, we used one-way ANOVA (function aov from package stats, v3.6.2) and a subsequent Pairwise t-test (function pairwise.t.test from package stats, v3.6.2), when normally distributed and showing homogeneous variance. For not normally distributed groups or showing a non-homogeneous variance, we used Kruskal Wallis test (function klustal.test from package stats, v3.6.2) and a subsequent Pairwise Wilcoxon rank sum exact test (function pairwise.woilcox.test from package stats, v3.6.2). The provided Source data file describes the statistical test and the resulting $p$-values for each plot.

## Reporting summary

Further information on research design is available in the Nature Portfolio Reporting Summary linked to this article.

## Data availability

Source data for plots in Figs. 1, 2, 4, 5 and Supplementary Figs. 2, 4, 7, 8 are provided with this paper in the Source data file. All the microscopy images generated for this study were deposited in BioImage Archive (accession number: S-BIAD1800). The FASTQ files from RNA sequencing in WT, *Hvclv1-1* and *Hvfcp1-1* inflorescences were deposited in the Sequence Read Archive (SRA) repository (BioProject: PRJNA1245663 [http://www.ncbi.nlm.nih.gov/bioproject/1245663]). Additional details are available from the corresponding author upon request. Source data are provided with this paper.

## Code availability

The custom Python code used for the phylogenetic tree construction is available on GitHub (https://github.com/Thilanka-lt/Pfam_domain_tree_construction).

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

## Acknowledgements

We thank Dr. Sebastian Hänsch and the Centre for Advanced imaging (CAi) at HHU for microscopy support and Dr. Tom Boissonnet for establishing the method used to measure HvCLV1 protein internalisation. We thank Ksenia Krooß for help with metadata annotation and upload to the BioImage Archive repository. We thank Edelgard Wendeler for technical support with barley transformation (MPIPZ), Meik Thiele for providing help with the statistical analysis, Karine Gustavo Pinto and Obada Chaar for their help with plant phenotyping. Work in R.S. and M.v.K.S. labs was supported by the DFG through CEPLAS (EXC2048), CSCS (FOR5235), NEXT-PLANT (IRTG2466) and work in I.F.A. lab by the Max Planck Society.

## Author contributions

I.V. performed most of the work presented here. J.E.M helped with the plant screening and imaging for the complementation of *Hvclv1-1* with the HvCLV1 reporter line. E.D.A. helped with the smRNA-FISH and with planning the experiments. T.L. from M.v.K.'s lab performed the RNA-seq raw data analysis and crossed the HvCLV1 reporter line into the barley *cv.* Golden Promise Fast. G.K.K cloned the vectors for HvCLV1 and HvFCP1 reporter lines, transformed by J.I. in barley cv. Golden Promise. I.F.A. helped with the transformation of *Hvclv1* mutants and K.M. in G.H.'s lab performed the transformation of *Hvfcp1* mutants. T.R. from S.S.'s lab performed the phylogenetic analysis, T.S. grew and analysed the *Hvclv1* mutant phenotype in semi-field-like conditions and R.S., with the cooperation of M.v.K., supervised the project and contributed to the overall idea and planning of the experiments.

## Funding

## Competing interests

The authors declare no competing interests.
