## [Transparent Peer Review file · Nature Communications]

CLAVATA signalling shapes barley inflorescence by controlling activity and determinacy of shoot meristem and rachilla

Corresponding Author: Professor Rüdiger Simon

Version 0:

Reviewer comments:

Reviewer #1

(Remarks to the Author)

The manuscript by Vardanega et al entitled "CLAVATA signaling shapes barley inflorescence architecture by controlling activity and determinacy of shoot apical and rachilla meristems" uncovers components of CLAVATA signaling in barley shoot meristems.

Starting with phylogenetic analysis, the authors identified HvCLV1, the closest ortholog of AtCLV1; they created 3 independent alleles of loss-of-function mutants using CRISPR-Cas9 genome editing. An impressive effort of creating multiple mutants, mutant complementation, reporter lines in this monocot crop plant deserves special appreciation. Well illustrated and explained barley inflorescence meristem components greatly help to understand the results. Strong sides of this work also include high quality imaging of the inflorescence meristems. This work is an excellent evidence confirming that CLAVATA pathway is a highly conserved pathway in land plants. On the mechanistic level, the authors show that close ortholog of CLV3 peptide gene in barley, HvFCP1 is strongly expressed in the vegetative meristem, later on in IM and in RM and it possibly acts via HvCLV1. In addition, authors performed transcriptomics analysis of wild type and mutant inflorescences at W3.5 and found a number of similarly up-regulated or down-regulated genes, suggesting a common targets of HvCLV1 and HvFCP1. Important to add, the authors use a large range of techniques to visualize proteins and gene expression patterns, from single-molecule RNA fluorescent in situ hybridization, to transcriptional and translational reporter lines

Coming to the weak points of this work. The novelty of these findings in the light of CLAVATA pathway nicely characterized in maize, *Setaria*, rice and many other species, is a bit questionable. The loss-of-function mutant *Hvclv1* has a mild phenotype and the loss-of-function mutant *Hvfc1* has even a weaker phenotype. The authors could create a higher order mutant, by removing additional RLKs or CLEs. The phylogenetic analysis of the CLE genes would be very important here. Overall, these barley mutants very much resemble phenotypes of *Setaria viridis* mutants published in 2021 in *Frontiers in Plant Sciences*. The current manuscript on barley does not offer any novel component of CLAVATA pathway, but rather confirms the conserved mechanism depending on auxin, cell division and T6P. Last findings come from the transcriptomics analysis and are not supported by additional functional tests. In conclusion, this is a nice work, bringing to the community new resources, confirming previously described pathway in the additional monocot plant.

The manuscript text has a number of typos and mistakes that need to be corrected; the abstract and the introduction need to be revised to better introduce the reader to the topic. In addition, citations need to be revised; I found often the original work showing an important insight is not cited (please see below the examples). The statistical tests performed in the data analysis are not explained well and it is not clear to me what do the 4 asterisks on the graphs mean?

Here are my more detailed comments:

-In the Abstract, line 33. There is a mistake, it should be CLAVATA3/EMBRYO-SURROUNDING REGION RELATED. Here the authors have to state already that CLAVATA pathway is known to regulate inflorescence architecture in grasses.

-In the Introduction, the authors explain in detail how the inflorescence is formed; it would be helpful if authors present in more detail what is known about CLAVATA pathway in inflorescence architecture in monocots and what is not known?

-Line 79: citation is missing. Please check carefully

-Line 128: the authors speculate that more pronounced protein internalization of CLV1 in RM indicates intracellular trafficking, or turnover etc. However, they do not quantify this phenomenon and it is difficult to judge such a statement.

-The complementation of *Hvclv1* is not quantified, it is difficult to rely on only one image. In addition, only the overall growth is presented, the other phenotypes need to be carefully tested.

-Regarding the phenotypes of the Hvclv1 mutants. It is clear that the mutant plants are shorter (F2A-the quantification is missing) and this is the most visible macrophenotype. It would be nice to test to what extent HvCLV1 contributes to vegetative growth and to what extent to the reproductive meristems. The authors restrict their focus on IM and do not explore enough the vegetative meristem.

-The most striking phenotype in inflorescence is the presence of multi-floret spikelets. The authors used SEM to identify the origin of this phenotype (Figure 3); it seems that in the absence of HvCLV1, the SRM identity stays longer

-Line 133: the authors claim clv1 mutant shows...fewer tillers, which is not true for all the alleles.

-Line 172: word "downregulated" is not correctly used, please replace

-Line 209: comma is missing before "respectively"

-Line 215: "Commonly" is not correctly used here. Please rephrase to have the correct message.

-Line 224: T6P should be used after introducing the abbreviation

-Line 252: In discussion, the authors claim that their "results point....in response to internal or external signals". It is not fully correct, they did not test any external signals.

-Line 272: the sentence is confusing, please rephrase to make the message clear. How the determinacy was shown in this work?

In the figure legends: sometimes authors use COLOR and sometimes COLOUR. Be consistent.

The definition of asterisks is missing in the figure legends.

Figure 6 is not mentioned in the text.

"asterisks" is often misspelled in the text.

Fig6 figure legends: the word "commonly" is not used correctly, better to be replaced by "similarly".

Thank you.

Reviewer #2

(Remarks to the Author)

Here, Vardanega et al. investigate the molecular regulation of inflorescence meristem maintenance and determinacy in barley by analysing the CLV1/3 signalling pathway. In many respects, this paper presents a paradigm shift in the technical capabilities that we expect from a cereal biology paper - the techniques used to analyse gene and protein expression with a diverse set of genetic resources and molecular approaches represent a major advance for cereal research. And while on the conceptual level it could be argued that the advance in understanding provided by this study is not so substantial given the knowledge known in Arabidopsis and maize, the question posed by the authors about how meristem identities are coordinated through inflorescence development is significant for the field, especially for major crops such as barley (and its closely related wheat) for which we know almost nothing about the CLV/WUS signalling pathway.

In general, I thought the paper is really well written and the conclusions are well supported by the data. There is a strong narrative that should make it easily understood by non-specialists, and all of the results are presented in a logical manner. The figures are fantastic, and the model at the end helps summarise the findings nicely.

I do however have a few major concerns and editorial suggestions for the authors to consider:

Major concerns

1) In some respects this point may be more about semantics than biology, but I don't really agree with the term "rachilla meristem", as it implies that this meristem is destined to form a rachilla, just like a spikelet meristem will form a spikelet or a floret meristem a floret. However, these group of cells don't seem to form a rachilla but instead are the apex or top of a lateral meristem. An alternative description for these cells could more simply be a lateral or axillary meristem from which the spikelet meristems are borne, or perhaps it is analogous to a branch meristem from species with panicle inflorescences. While I agree with the authors that there is a meristem like mound of cells that sits on top of the spikelet primordia, which should have a name, I don't really see why they should be considered to be a rachilla primordia - would this not be like calling the IM a Rachis Primordium? I believe the authors are introducing this term as I have not seen it described so anywhere else, and consequentially, I think it should be justified better than the current description on L157-161.

2) Many results examine meristem size phenotypes, including width, height, volume, area etc. However, I cannot find anywhere how these measurements were determined. Could the authors please provide more details about the software, code, or scripts used to obtain these measurements? I am also curious about the method used to standardise measurements between genotypes and samples - are the heights and widths taken at a point 50 or 100 um from the apex of the IM? Or were they taken at the first sign of an emerging lateral meristem? More detail is needed here.

3) It is a little disappointing that the results of this paper are not discussed alongside those of Wang et al., who investigated FASCIATED EAR genes that include CLV-like genes, especially because Wang et al. corroborate, at least in part, the results shown here that a bigger meristem does not mean a higher yielding plant in barley. Is there a reason Wang et al. has not been included in the discussion, for example?

4) Similarly, while I appreciate the clarity of the RNA-seq results and the link to inflorescence branching pathways, I have the feeling it might be too simplistic. For example, in Wang et al., the authors identified a number of MADS-box and AP2

transcription factors that were differentially expressed. Are any of these genes mis-regulated in the *clv1* or *fcp1* mutants given they exhibit spikelet and floret architecture phenotypes? Also, many of the mentioned genes share similarities with those that are mis-regulated in the *VRS4* mutant (paper published by Schnurbusch's group) - are there any similarities in the mis-regulated genes between *clv1/fcp1* and *vrs4* mutants?

Some minor comments

- 1) In Figure 1C, is the signal detected in leaves and at the base of the stem background signal, or do you the authors think it is true signal? If so, the broader signal should be mentioned in the text.
- 2) In a number of graphs, there is no y-axis label provided, which could confuse readers. Examples include 2D, 2G, and multiple panels in extended figures 2 and 4. Please include labels.
- 3) Similarly, in graphs where genotypes are provided on the x axes, the genotypes should be written in italics.
- 4) In Figure 4A, why have the authors reverted to using SAM rather than IM? I recommend being consistent. And, in this graph, the authors have used box and violin plots, where as box plots are used in every other figure. Why is this so? Why include a box and violin plot together?
- 5) Figure 6A - correct the spelling of Trehalose.
- 6) L113: change "similarly" to "similar".
- 7) L114-121: this is a really long sentence, consider splitting into two.
- 8) L125: change "flower organs" to "floral organs"
- 9) L233: Maybe "Consistent with" rather than "Consistently"
- 10) L245: correct to "as a consequence"

Version 1:

Reviewer comments:

Reviewer #1

(Remarks to the Author)

Reviewer #2

(Remarks to the Author)

I thank the authors for the very considered responses to my comments. The revised version that contains valuable new data has addressed all of my concerns.

I congratulate the authors on an excellent study filled with great experimental data and new insights for the field. I recommend it be published in Nature Comms.

REVISIONS

Reviewer #1 (Remarks to the Author)

The manuscript by Vardanega et al entitled "CLAVATA signaling shapes barley inflorescence architecture by controlling activity and determinacy of shoot apical and rachilla meristems" uncovers components of CLAVATA signaling in barley shoot meristems. Starting with phylogenetic analysis, the authors identified HvCLV1, the closest ortholog of AtCLV1; they created 3 independent alleles of loss-of-function mutants using CRISPR-Cas9 genome editing. An impressive effort of creating multiple mutants, mutant complementation, reporter lines in this monocot crop plant deserves special appreciation. Well illustrated and explained barley inflorescence meristem components greatly help to understand the results. Strong sides of this work also include high quality imaging of the inflorescence meristems. This work is an excellent evidence confirming that CLAVATA pathway is a highly conserved pathway in land plants. On the mechanistic level, the authors show that close ortholog of CLV3 peptide gene in barley, HvFCP1 is strongly expressed in the vegetative meristem, later on in IM and in RM and it possibly acts via HvCLV1. In addition, authors performed transcriptomics analysis of wild type and mutant inflorescences at W3.5 and found a number of similarly up-regulated or down-regulated genes, suggesting a common targets of HvCLV1 and HvFCP1. Important to add, the authors use a large range of techniques to visualize proteins and gene expression patterns, from single-molecule RNA fluorescent in situ hybridization, to transcriptional and translational reporter lines

Thank you for the appreciation of our work.

Coming to the weak points of this work. The novelty of these findings in the light of CLAVATA pathway nicely characterized in maize, Setaria, rice and many other species, is a bit questionable.

The term "CLAVATA pathway" is not precisely defined, but relates to findings in Arabidopsis, where the CLE peptide CLV3 and the key receptor CLV1 control the size of the stem cell domain in interaction with the underlying organising centre. In maize, rice, Setaria and other species (tomato!), related signalling pathways involving CLE peptides and CLV1-like receptors have been found to also control inflorescence and/or floral meristems sizes. However, further studies in Arabidopsis uncovered the meristem activities of additional CLE peptides (such as CLE40) and CLV1-related receptors (BAM1, others), indicating that meristem function and development are shaped by multiple, potentially independently acting CLE - receptor signalling pathways.

In maize, the CLV-related pathways restrict inflorescence and floral meristem size, similar to the findings made for Arabidopsis. In rice, the RLK FON1 and the CLV3-like peptide FON2 affect the number of floret organs, and the CLE-peptides FCP1 and FCP2 are both required to maintain

vegetative meristem activity. Interaction between FON1 and FCP1 was tested in rice vegetative meristems, but excluded (now added in lines 93-95). The *Setaria* CLV3 orthologue *SvFON2* confines inflorescence and branch meristem activity, and *Svfon2* mutant meristems fasciate ("fasciation" is a rather loose term to describe proliferation of meristem cells of uncharacterised identity).

We show here that in barley, and in contrast to the findings in other grasses, a pathway comprising the CLE peptide *HvFCP1* and the receptor *HvCLV1* regulates and coordinates the activity of a subset of meristem types within the inflorescence. The phenotypes we observed in the barley *Hvclv1* and *Hvfcp1* mutant differ from those found in other cereal species: we here describe a prolonged activity of the inflorescence meristem and the rachilla primordium, which ultimately affects the number of florets produced on the main axis and at lateral positions. In addition, the formation of additional rows of spikelets in the *Hvclv1* "crowned spikes" is the result of an organised change in the spikelet phyllotaxis, rather than fasciation as was described in other grasses. The additional phenotypes we observed for *Hvclv1* mutants, compared to *Hvfcp1*, showed that *HvCLV1* can perceive different inputs, i.e. peptide signals during inflorescence development. These results differ from previous findings for other grasses, and support our hypothesis that several different CLE-peptide dependent pathways regulate the activity of specific meristem types within the complex grass inflorescences.

The loss-of-function mutant *Hvclv1* has a mild phenotype and the loss-of-function mutant *Hvfcp1* has even a weaker phenotype. The authors could create a higher-order mutant, by removing additional RLKs or CLEs.

We wrote in the manuscript that the phenotypes of *Hvclv1* or *Hvfcp1* appear mild, compared to the strongly fasciated meristems observed in maize *td1* or *Setaria Svfon2* mutants (notice that *Svfon2* is not the closest orthologue to *Hvfcp1*). The non-fasciation of our barley mutants indicates that the *HvFCP1/HvCLV1* pathway executes a very distinct activity to confine inflorescence meristem growth, promote SM formation and, importantly, restrict growth of the rachilla. Studying other CLE peptides their receptors, the signalling pathways that they establish and their specific roles in barley development is certainly of high interest to us, but does not necessarily add to our understanding of the *FCP1/CLV1* pathway itself.

The phylogenetic analysis of the CLE genes would be very important here.

Thank you for the suggestion, we provided a list of all the CLE peptides in barley in Supplementary Figure 5. We didn't include a phylogenetic tree because performing a phylogenetic analysis on such short protein sequences (12aa) carries only very limited information, and each individual branch of the phylogenetic tree would be inevitably not significant. On the other end, we realised it was necessary to include additional data clarifying that *HvFCP1* is not the closest ortholog of *CLV3* in barley, and to show that a *CLV3* ortholog exists in barley and was not investigated in this study (Supplementary Figure 5 B,C).

Overall, these barley mutants very much resemble phenotypes of *Setaria viridis* mutants published in 2021 in *Frontiers in Plant Sciences*.

The component of the CLAVATA pathway characterised in *Setaria viridis* in the cited publication is SvFON2, the closest ortholog of the Arabidopsis CLE peptide CLV3 and rice FON2. In the same paper, it is reported that: “Sequence comparisons indicate that FCP1 gene sequences are conserved among grasses (Supplementary Figure 2B), consistent with the hypothesis that the function of SvFCP1 is similar to that of ZmFCP1 and are likely not central to grass inflorescence development”. This sentence is based on a hypothesis, which is in contrast to the novel results that we report here. To clarify: The *Setaria* SvFCP1 peptide exists and shares an identical sequence with HvFCP1, which is different from the sequence of SvFON2. Additionally, we identified the closest ortholog to the Arabidopsis CLV3 and the *Setaria* FON2 peptide in barley and named it HvCLV3. However, HvCLV3 was not characterised in this work.

The role of the *Setaria* FON2 in the regulation of inflorescence and branch meristems size, together with the results from Arabidopsis indicating an interaction between CLV3 and CLV1, only suggests that the barley ortholog of FON2 (HvCLV3) could be a possible additional interactor of HvCLV1, in line with our results, suggesting that additional CLE peptides could interact with HvCLV1.

The current manuscript on barley does not offer any novel component of CLAVATA pathway, but rather confirms the conserved mechanism depending on auxin, cell division and T6P. Last findings come from the transcriptomics analysis and are not supported by additional functional tests.

Thank you for your comment. Yes, we didn't offer any novel component of CLAVATA pathway, but we identified interaction between HvFCP1 and HvCLV1, which was previously rejected in rice (lines 93-95) and never tested in other species. We added results from smRNA-FISH in WT and *Hvclv1* inflorescences (Supplementary Figure 10) to support results from the RNA-seq analysis. Even though the connection between T6P and inflorescence branching is already known, the comment implies that the link between the CLAVATA pathway and T6P signalling was firmly demonstrated. Based on the finding that *Setaria* RA3 is in a co-expression network with SvFON2. Here, we show upregulation of TPS and downregulation of SRA, in addition to upregulation of genes affected by T6P levels as FT2 and SWEET11 in both *Hvclv1* and *Hvfcv1* mutants, strongly supporting the notion that HvTPS1 and HvSRA are targets of the HvFCP1/HvCLV1 signalling pathway. We feel that these results significantly extend previous work performed in other species.

In conclusion, this is a nice work, bringing to the community new resources, confirming previously described pathway in the additional monocot plant. The manuscript text has a number of typos and mistakes that need to be corrected; the abstract and the introduction need to be revised to better introduce the reader to the topic. In addition, citations need to be revised; I found often the original work showing an important insight is not cited (please see below the examples).

Thank you for your comment, we implemented some additional information regarding the role of CLAVATA-related genes in different grasses and included the main phenotypes observed in *Setaria Viridis fon2* (lines: 88-112)

The statistical tests performed in the data analysis are not explained well and it is not clear to me what do the 4 asterisks on the graphs mean?

Thank you for your comment, we better clarified the performed statistical analysis in the material and methods section (lines: 512-528). Additionally, we included all the details about the statistical test and the resulting p-value for each plot in the Source Data file. We added the statistical test performed and the asterisk description in each figure legend.

Here are my more detailed comments:

-In the Abstract, line 33. There is a mistake, it should be CLAVATA3/EMBRYO-SURROUNDING REGION RELATED. Here the authors have to state already that CLAVATA pathway is known to regulate inflorescence architecture in grasses.

Thank you for your comment, we added the required sentence to the abstract (line 28).

-In the Introduction, the authors explain in detail how the inflorescence is formed; it would be helpful if authors present in more detail what is known about CLAVATA pathway in inflorescence architecture in monocots and what is not known?

Thank you for your comment, we added more on what is known and what is not known about the regulation of inflorescence architecture by CLAVATA genes from line 77 to 112 of the introduction.

-Line 79: citation is missing. Please check carefully

Thank you for your comment, citations were added (line 79).

-Line 128: the authors speculate that more pronounced protein internalization of CLV1 in RM indicates intracellular trafficking, or turnover etc. However, they do not quantify this phenomenon and it is difficult to judge such a statement.

Thank you for the suggestion, we used an image-based analysis using the software Fiji to quantify *Hvclv1* internalisation in RP and FM in WT and *Hvfc1* mutant background. The method used is now described in the material and methods section (lines: 374-387) and in Supplementary Figure 1D, the results were added to Figure 1 and Supplementary Figure 8E.

-The complementation of *Hvclv1* is not quantified, it is difficult to rely on only one image. In addition, only the overall growth is presented, the other phenotypes need to be carefully tested.

Thank you for your comment, we measured plant, inflorescence and grain phenotypes of the complemented line in comparison to WT and *Hvclv1* (Supplementary Figure 4).

-Regarding the phenotypes of the *Hvclv1* mutants. It is clear that the mutant plants are shorter (F2A- the quantification is missing) and this is the most visible macrophenotype.

Thank you for your comment, all the quantifications related to Figure 2A are shown in Supplementary Figure 2 B-K. Including quantification of stem length which clearly describes the shortness of *Hvclv1* plants.

- It would be nice to test to what extent HvCLV1 contributes to vegetative growth and to what extent to the reproductive meristems. The authors restrict their focus on IM and do not explore enough the vegetative meristem.

We provided measurements of the *Hvclv1* SAM in comparison to WT at the vegetative stage in Figure 4A. We changed the name to vSAM in lines 198-203 to better clarify the measurements were referred to the vegetative meristem.

-The most striking phenotype in inflorescence is the presence of multi-floret spikelets. The authors used SEM to identify the origin of this phenotype (Figure 3); it seems that in the absence of HvCLV1, the SRM identity stays longer.

Thank you for your comment, we got to the same conclusions.

-Line 133: the authors claim *clv1* mutant shows...fewer tillers, which is not true for all the alleles.

Thank you for your comment, we clarified this in line 155.

-Line 172: word "downregulated" is not correctly used, please replace.

Thank you for your comment, we changed the wording (now in line 203).

-Line 209: comma is missing before "respectively"

Thank you for your comment, we added the comma (now in line 241).

-Line 215: "Commonly " is not correctly used here. Please rephrase to have the correct message.

Thank you for your comment, we replaced "commonly" with "similarly" (now in line 247).

-Line 224: T6P should be used after introducing the abbreviation

Thank you for your comment, the abbreviation "T6P" is now used only after being introduced in line 276.

-Line 252: In discussion, the authors claim that their "results point....in response to internal or external signals". It is not fully correct, they did not test any external signals.

Thank you for your comment, as suggested, we removed “external signals” (now in line 303).

-Line 272: the sentence is confusing, please rephrase to make the message clear. How the determinacy was shown in this work?

Thank you for your comment, we clarified that the formation of additional florets suggests a change in determinacy in lines 295-297 of the discussion.

In the figure legends: sometimes authors use COLOR and sometimes COLOUR. Be consistent. The definition of asterisks is missing in the figure legends. Figure 6 is not mentioned in the text.

"asterisks" is often misspelled in the text.

Thank you for your comment, we changed the misspelled "asterisks" and replaced “color” with “colour” throughout the text.

Fig6 figure legends: the word "commonly" is not used correctly, better to be replaced by "similarly".

Thank you for your comment, we followed your suggestion.

Thank you.

Reviewer #2 (Remarks to the Author):

Here, Vardanega et al. investigate the molecular regulation of inflorescence meristem maintenance and determinacy in barley by analysing the CLV1/3 signalling pathway.

comment: we analysed the HvFCP1/HvCLV1 pathway. HvCLV3 was not analysed in this paper.

In many respects, this paper presents a paradigm shift in the technical capabilities that we expect from a cereal biology paper - the techniques used to analyse gene and protein expression with a diverse set of genetic resources and molecular approaches represent a major advance for cereal research. And while on the conceptual level it could be argued that the advance in understanding provided by this study is not so substantial given the knowledge known in Arabidopsis and maize, the question posed by the authors about how meristem identities are coordinated through inflorescence development is significant for the field, especially for major crops such as barley (and its closely related wheat) for which we know almost nothing about the CLV/WUS signalling pathway.

In general, I thought the paper is really well written and the conclusions are well supported by the data. There is a strong narrative that should make it easily understood by non-specialists, and all of the results are presented in a logical manner. The figures are fantastic, and the model at the end helps summarise the findings nicely.

I do however have a few major concerns and editorial suggestions for the authors to consider:

Major concerns

1) In some respects this point may be more about semantics than biology, but I don't really agree with the term "rachilla meristem", as it implies that this meristem is destined to form a rachilla, just like a spikelet meristem will form a spikelet or a floret meristem a floret. However, these group of cells don't seem to form a rachilla but instead are the apex or top of a lateral meristem. An alternative description for these cells could more simply be a lateral or axillary meristem from which the spikelet meristems are borne, or perhaps it is analogous to a branch meristem from species with panicle inflorescences. While I agree with the authors that there is a meristem like mound of cells that sits on top of the spikelet primordia, which should have a name, I don't really see why they should be considered to be a rachilla primordia - would this not be like calling the IM a Rachis Primordium? I believe the authors are introducing this term as I have not seen it described so anywhere else, and consequentially, I think it should be justified better than the current description on L157-161.

Thank you for your comment. We agree we can't provide any evidence supporting the idea that the region we marked as RP in Figure 1 is going to develop the rachilla, we therefore removed Figure 1J and substituted the term "rachilla meristem" with "rachilla primordium", consistently with the description provided by 'Koppolu et al. 2021' (now cited). The rachilla primordium (RP) become distinct from the spikelet meristem only after definition of the floret meristem, we therefore marked the RP only in the mentioned stage in Figure 5A. Additionally, we added scanning electron microscope pictures of the mature rachilla in WT and *Hvclv1-1*, describing its phenotype (lines 186-190, Supplementary Figure 3C). We think the term "rachilla primordium" is the one which better defines such organ, and we preferred to adopt this already published term instead of using an additional name which would possibly generate confusion.

2) Many results examine meristem size phenotypes, including width, height, volume, area etc. However, I cannot find anywhere how these measurements were determined. Could the authors please provide more details about the software, code, or scripts used to obtain these measurements? I am also curious about the method used to standardise measurements between

genotypes and samples - are the heights and widths taken at a point 50 or 100 um from the apex of the IM? Or were they taken at the first sign of an emerging lateral meristem? More detail is needed here.

Thank you for your comment. We now clarified in the figure legends that the IM measurements in Figure 2K and the new Supplementary Figure 7 F,G were taken drawing a horizontal line from the last emerging spikelet meristem at a defined W stage (IM width) and a perpendicular vertical line connecting it to the highest IM point (IM height). We added segmented white lines in Figure 2K to better clarify how the measurements were taken. For the calculation of IM volumes in Figure 4 I,J, we considered all the cells above a horizontal line traced from the last emerging spikelet meristem.

3) It is a little disappointing that the results of this paper are not discussed alongside those of Wang et al., who investigated FASCIATED EAR genes that include CLV-like genes, especially because Wang et al. corroborate, at least in part, the results shown here that a bigger meristem does not mean a higher yielding plant in barley. Is there a reason Wang et al. has not been included in the discussion, for example?

Thank you for your comment, we now added Wang et al. to the reference list and discussed it in correlation with our results in lines 304-306.

4) Similarly, while I appreciate the clarity of the RNA-seq results and the link to inflorescence branching pathways, I have the feeling it might be too simplistic. For example, in Wang et al., the authors identified a number of MADS-box and AP2 transcription factors that were differentially expressed. Are any of these genes mis-regulated in the *clv1* or *fcp1* mutants given they exhibit spikelet and floret architecture phenotypes? Also, many of the mentioned genes share similarities with those that are mis-regulated in the *VRS4* mutant (paper published by Schnurbusch's group) - are there any similarities in the mis-regulated genes between *clv1/fcp1* and *vrs4* mutants?

Thank you for your comment. We now added some more details from our RNA-seq results. Even though we found two MADS-box proteins (*HvMADS1* and *HvMADS3*) to be downregulated in *Hvclv1* and *Hvfcp1* vs WT, smRNA-FISH results indicated that their downregulation was probably linked to the reduced number of spikelet primordia in both mutants. Interestingly, ectopic expression of *HvMADS1* in the IM could be relevant for the observed phenotype. We added smRNA_FISH results also for *HvIAA31* and *HvMND6*, and we found them to be interestingly upregulated in the main rachis and RP of *Hvclv1* (Supplementary Figure 10). Additionally, we considered important to mention the upregulation of *HvBAM5* in *Hvclv1*, a LRR-RLK part of the same clade as *HvCLV1* (Supplementary fig. 1A) which can possibly partially rescue the *Hvclv1* phenotype. Moreover, we mentioned in the text (lines 317-321) the similarities in the regulation of T6P signalling with the barley mutants *vrs4* and *com1*. *VRS4* and *COM1* were not significantly

differentially expressed in *Hvclv1* or *Hvfcp1*, even though downregulation of *COM2* in *Hvclv1* vs WT suggests a possible shared pathway with HvCLV1 in the regulation of T6P levels.

Some minor comments

1) In Figure 1C, is the signal detected in leaves and at the base of the stem background signal, or do you the authors think it is true signal? If so, the broader signal should be mentioned in the text.

Thank you for your comment. We mentioned the HvCLV1 expression in leaf primordia in line 142.

2) In a number of graphs, there is no y-axis label provided, which could confuse readers. Examples include 2D, 2G, and multiple panels in extended figures 2 and 4. Please include labels.

Thank you for your comment. Labels were added in the y-axis of each plot.

3) Similarly, in graphs where genotypes are provided on the x axes, the genotypes should be written in italics.

Thank you for your comment. Mutant genotypes in the x-axes are now written in italics.

4) In Figure 4A, why have the authors reverted to using SAM rather than IM? I recommend being consistent. And, in this graph, the authors have used box and violin plots, where as box plots are used in every other figure. Why is this so? Why include a box and violin plot together?

Thank you for your comment. We used SAM because measurements were taken in meristems at the vegetative stage, therefore not in IM. To further clarify we now used the abbreviation vSAM and mentioned more clearly the stage in which the measurements were taken in the text (lines 198-203). The violin plots in Figure 4A were removed and a grey background was used instead to distinguish the treated samples from the mock samples (white background).

5) Figure 6A - correct the spelling of Trehalose.

Thank you, fixed.

6) L113: change "similarly" to "similar".

Thank you, fixed (line 130).

7) L114-121: this is a really long sentence, consider splitting into two.

Thank you, fixed (line 134).

8) L125: change "flower organs" to "floral organs"

Thank you, fixed (line 141).

9) L233: Maybe "Consistent with" rather than "Consistently"

Thank you, fixed (line 285).

10) L245: correct to "as a consequence"

Thank you, fixed (line 294).